# The global importance and interplay of colour-based protective and thermo-regulatory functions in frogs

Ricarda Laumeier [1,2] ✉, Martin Brändle[1], Mark-Oliver Rödel[3], Stefan Brunzel[2], Roland Brandl[1] & Stefan Pinkert [4,5]

Small-scale studies have shown that colour lightness variation can have important physiological implications in ectotherms, with darker species having greater heating rates, as well as protection against pathogens and photo-oxidative damage. Using data for 41% (3059) of all known frog and toad species (Anura) from across the world, we reveal ubiquitous and strong clines of decreasing colour lightness towards colder regions and regions with higher pathogen pressure and UVB radiation. The relative importance of pathogen resistance is higher in the tropics and that of thermoregulation is higher in temperate regions. The results suggest that these functions influence colour lightness evolution in anurans and filtered for more similarly coloured species under climatic extremes, while their concurrent importance resulted in high within-assemblage variation in productive regions. Our findings indicate three important functions of colour lightness in anurans – thermoregulation, pathogen and UVB protection – and broaden support for colour lightness-environment relationships in ectotherms.

In the face of climate change and biodiversity decline, trait-based inferences of the processes that underpin species' distributions are becoming increasingly important because of their potential for understanding and forecasting biological responses[1–3]. The most fundamental and general explanations for reaction norms across taxa, biogeographical realms, and spatial scales—so-called ecogeographical rules—address morphological features with broad impacts on species' development, activity, and reproduction[4,5]. Ecogeographical research was inspired by evidence for size-based thermoregulation in mammals and birds[6]. Consequently, body size is by far the most commonly used morphological predictor for determining species' responses to climate change[2,3]. However, size-based thermoregulation is less important in species that rely on external rather than metabolically produced heat

(i.e., ectotherms)[7,8], which constitute 99.9% of all animal species[9]. In addition, despite the crucial role of species' thermal requirements in models of range shifts and extinction risk, the effects of body size therein are often weak and inconclusive[2,3].

Colour variation plays a crucial role in species' biotic interactions, including aposematism and camouflage[10–12]. A growing body of macroecological and experimental studies suggest that colour lightness also has multiple physiological functions in insects, reptiles and birds, with important implications for species' distributions[13–16]. For instance, the biophysical principle that darker objects heat up faster than lighter ones confers a thermoregulatory advantage to darker-coloured species in colder environments (thermal melanism hypothesis, also known as Bogert's rule)[13,17–19]. At the same time, lighter-coloured species seem

[1]Department of Animal Ecology, Faculty of Biology, Philipps-Universität Marburg, Karl-von-Frisch-Straße 8, 35043 Marburg, Germany. [2]Department of Biodiversity and Species Conservation, Faculty of Landscape Architecture, Horticulture and Forestry, University of Applied Science Erfurt, Leipziger Straße 77, 99085 Erfurt, Germany. [3]Department of Evolutionary Diversity Dynamics, Museum für Naturkunde—Leibniz Institute for Evolution and Biodiversity Science, Invalidenstraße 43, 10115 Berlin, Germany. [4]Department of Ecology and Evolutionary Biology, Yale University, 165 Prospect Street, 06511 New Haven, CT, USA. [5]Department of Conservation Ecology, Faculty of Biology, Philipps-Universität Marburg, Karl-von-Frisch-Straße 8, 35043 Marburg, Germany. ✉e-mail: ricardalaumeier@gmail.com

to have an advantage in warmer environments due to enhanced reflection of light in the visible and infrared spectrum of the light that prevents overheating[16,20,21]. In addition, dark-coloured pigments, melanins, are also known to have protective functions. On the one hand, darker colours provide greater protection against UVB radiation[22–25]. On the other hand, melanins (the main colour pigments) enhance the structural integrity of cells and a higher melanin concentration in the cuticle is associated with enhanced immunocompetence, providing greater protection against penetration by fungal and bacterial pathogens e.g.[26–28]. Thus, darker species are assumed to have an advantage under warm and wet conditions where pathogens thrive particularly well (Gloger's rule[11,15,29,30]). Understanding these physiological functions of colour lightness is relevant not only in the context of climatic changes but also in the face of massive population declines and local extinctions caused by pathogens such as pandemic chytrid disease[31,32]. However, despite growing evidence for their ecological significance, support for the functions of colour lightness is generally limited in both taxonomic and spatial extent[13], and their interplay in structuring communities remains unknown.

Here, we investigated whether contemporary species distributions and the functional composition of assemblages are explained by colour-based thermoregulation, pathogen resistance and UVB protection using data for 41% of all species of frogs and toads (Order: Anura) from across the world. In line with the thermal melanisms hypothesis, Gloger's rule and the UVB-protection hypothesis that refer to the thermoregulatory and protective functions of colour lightness in animals, we expected an increase in the colour lightness of anurans with increasing temperature (mean annual temperature and decreasing elevation), decreasing productivity (enhanced vegetation index) and decreasing UVB radiation. We tested these main hypotheses at both the assemblage- and species-level. Next, we assessed the extent to which these functions shape the phylogenetic signal underlying contemporary colour lightness variation. Because regionally dominant environmental drivers might favour specific functions of colour lightness, we also explored their independent contributions across biogeographical realms as well as their impact on filtering for species more similar in colour lightness (i.e., functional diversity). Biological information for anurans is uniquely rich and complete compared to that for other ectothermic taxa, providing an opportunity not only to rigorously test our predictions but also to identify mechanisms that—from a physiological point of view—are readily applicable to most animal taxa.

Our study reveals that the colour lightness of anurans is consistently positively affected by temperature (thermal melanism hypothesis) and negatively affected by productivity (i.e. warm-wet conditions, Gloger's rule) as well as UVB irradiance (UV-protection hypothesis). Furthermore, we show that colour lightness is more similar between closely related species and that the phylogenetically predicted part of colour lightness is mostly driven by colour-based thermoregulation. This suggests that the evolution of colour lightness favoured the colonisation of temperate climates of a few closely related lineages of the Anura. Our global-scale analysis supports findings from smaller-scaled studies for other ectothermic taxa and consolidates the fundamental role of colour variation for the distribution and climate change responses of animal species.

## Results
### Global assemblage-level analyses
In models of the average colour lightness of assemblages (co-occurring species) and environmental factors, colour lightness increased with increasing temperature and elevation as well as decreasing productivity and decreasing UVB radiation. Temperature and productivity were the most important predictors of this pattern (Table 1, Fig. 1). All variables together explained 59% of the variation in colour lightness in models that included a geographical trend

surface term to account for latent spatially autocorrelated variables (Table 1).

To assess the extent to which the putative function influenced the distribution of anuran lineages and the biogeographical patterns underlying colour lightness variation, we decomposed the raw colour lightness of anuran species into the part predicted by the phylogeny (the phylogenetic signal) and its phylogenetically independent deviation from this prediction (species-specific signal) using Lynch's comparative method[33]. Subsequently, we averaged these two trait components across co-occurring species for assemblage-level analyses. Separate models for the phylogenetic and species-specific components of colour lightness of assemblages showed that the phylogenetic component was mainly predicted by temperature and UVB radiation (colour-based thermoregulation and UVB protection), whereas the phylogenetically independent component was mainly correlated with productivity (pathogen resistance; Table 1). All variables together explained 70% of the variation in the phylogenetically predicted and 52% in the species-specific part of colour lightness in generalised additive models (GAM) that included a geographical trend surface term to account for latent spatially autocorrelated variables (Table 1, for single linear regressions, see Supplementary Fig. 1).

### Realm-specific differences in the independent contributions of functions
When accounting for spatial autocorrelation in assemblage-level models through trend surface GAM, the effects of environmental drivers were generally weaker but remained robust and did not change in their direction or relative importance (Table 1). The trend surface term of longitude and latitude in these models explained a rather large additional proportion of the variance in colour lightness compared to spatially naive models (GAM vs. LM: 59% > 24%), suggesting that these global models lacked a spatially structured latent variable. However, interaction terms of the environmental variables and the biogeographical realm as predictors explained 52% of the variation in the colour lightness of anuran assemblages (Fig. 2). Hence, accounting for idiosyncrasies of the responses among realms with different biogeographical histories largely removed the effect of latent spatially autocorrelated variables.

Across biogeographical realms, the effects of environmental variables on the average colour lightness of anuran assemblages were largely consistent but differed in magnitude (Fig. 2). Of the 13 biogeographical regions with more than 50 grid cells covered by our data, eight showed positive effects of temperature (mean annual temperature or elevation). Productivity negatively affected colour lightness in six realms (Fig. 2). Hierarchical partitioning analyses of the independent effects of environmental predictors on colour lightness showed that proxies for temperature were the dominant predictors of the variation in assemblage-level colour lightness in five of six temperate realms as well as Madagascar and Sub-Saharan Africa. Productivity (i.e. mean annual enhanced vegetation index; EVI) was most important in three of the seven tropical realms and mean annual UVB radiation was most important in northern South America, the Australis, and the Western Palaearctic.

Colour lightness variation among co-occurring species, i.e., colour lightness diversity (see Methods), generally decreased with increasing mean annual UVB radiation and increased with increasing mean annual temperature, elevation and productivity (Table 1). Environmental predictors together explained 35% of the variation in colour lightness diversity in models that account for latent spatially autocorrelated variables. However, the cumulative and relative (predictor-specific) importance of environmental predictors differed between biogeographical realms. All variables together explained between 5% and 67% of the variation in colour lightness diversity among the 13 investigated biogeographical realms (Supplementary Fig. 2). Proxies for temperature were the most important drivers of colour lightness diversity in

**Table 1 | Environmental drivers of the colour lightness of anuran assemblages**

| Variable | | Predictor | Linear model | | | | Generalised additive model | | | |
|---|---|---|---|---|---|---|---|---|---|---|
| | | | Slope ± SE | t | P | $R^2$ | Slope ± SE | t | P | $R^2$ |
| Average colour lightness | Raw | MAT | **19.47 ± 0.46** | **42.76** | **<$10^{-16}$** | 0.24 | 9.89 ± 0.79 | 12.57 | <$10^{-16}$ | 0.59 |
| | | Elevation | 4.28 ± 0.17 | 24.58 | <$10^{-16}$ | | 2.10 ± 0.23 | 9.05 | <$10^{-16}$ | |
| | | EVI | −5.12 ± 0.14 | −35.48 | <$10^{-16}$ | | **−2.63 ± 0.18** | **−14.52** | **<$10^{-16}$** | |
| | | UVB | −7.84 ± 0.39 | −20.35 | <$10^{-16}$ | | −8.44 ± 0.66 | −12.70 | <$10^{-16}$ | |
| | P component | MAT | 1.59 ± 0.16 | 10.22 | <$10^{-16}$ | 0.08 | 2.70 ± 0.21 | 13.09 | <$10^{-16}$ | 0.70 |
| | | Elevation | −0.47 ± 0.06 | −7.80 | $6.51 \times 10^{-15}$ | | 0.32 ± 0.06 | 5.19 | $2.11 \times 10^{-07}$ | |
| | | EVI | **−1.07 ± 0.05** | **−21.63** | **<$10^{-16}$** | | −0.31 ± 0.05 | −6.52 | $7.18 \times 10^{-11}$ | |
| | | UVB | 0.10 ± 0.13 | 0.76 | $4.47 \times 10^{-01}$ | | **−2.95 ± 0.17** | **−16.94** | **<$10^{-16}$** | |
| | S component | MAT | **17.88 ± 0.36** | **49.28** | **<$10^{-16}$** | 0.27 | 7.20 ± 0.69 | 10.39 | <$10^{-16}$ | 0.52 |
| | | Elevation | 4.75 ± 0.14 | 34.19 | <$10^{-16}$ | | 1.79 ± 0.20 | 8.74 | <$10^{-16}$ | |
| | | EVI | −4.06 ± 0.12 | −35.24 | <$10^{-16}$ | | **−2.32 ± 0.16** | **−14.57** | **<$10^{-16}$** | |
| | | UVB | −7.94 ± 0.31 | −25.86 | <$10^{-16}$ | | −5.49 ± 0.58 | −9.40 | <$10^{-16}$ | |
| Colour lightness diversity | | MAT | 0.01 ± 0.03 | 0.41 | $6.83 \times 10^{-01}$ | 0.09 | 0.44 ± 0.07 | 6.69 | $2.31 \times 10^{-11}$ | 0.35 |
| | | Elevation | 0.14 ± 0.01 | 11.32 | <$10^{-16}$ | | **0.20 ± 0.02** | **10.62** | **<$10^{-16}$** | |
| | | EVI | **−0.12 ± 0.01** | **−12.77** | **<$10^{-16}$** | | 0.02 ± 0.01 | 1.76 | $7.81 \times 10^{-02}$ | |
| | | UVB | −0.24 ± 0.03 | −8.84 | <$10^{-16}$ | | −0.52 ± 0.05 | −9.82 | <$10^{-16}$ | |

Linear (LM) and generalised additive models (GAM) of environmental predictors of the raw colour lightness, its phylogenetic component (P), and its species-specific component (S) (n = 16,686 assemblages) as well as the colour lightness diversity of anuran assemblages (n = 14,436 assemblages). The significance of the correlation coefficients was tested using a two-sided t-test. The effect of species richness was controlled for in the colour-lightness diversity calculation (see Methods). Models of colour lightness diversity include fewer assemblages because singletons are invariant. GAMs additionally include a smoothed (trend surface) term of assemblage coordinates to account for spatial autocorrelation.
Note: *MAT* mean annual temperature, *EVI* mean annual EVI, *UVB* mean annual UVB. Predictors with the strongest effect per model are bold.

three of six temperate and five of seven tropical realms. Mean annual UVB radiation and temperature were similarly important in the Western Palaearctic. Annual UVB radiation was the most important predictor in North America, northern South America and southern South America. Productivity was the main predictor of colour lightness diversity in Central Tropical Africa.

## Species-level analyses

Pagel's lambda-based tests for phylogenetic signals in both colour lightness and averages of the environmental factors across species' ranges showed a strong impact of phylogenetic relatedness in our data ($\lambda_{Colour}$ = 0.49, $\lambda_{Temperature}$ = 0.88, $\lambda_{Elevation}$ = 0.86, $\lambda_{Productivity}$ = 0.85, $\lambda_{UVB}$ = 0.88). At the species level, we, therefore, explored the drivers of colour lightness with models that included interactions between environmental drivers and anuran families. In these models, species of families with fewer than 10 species were excluded, which reduced the data to 2984 species from 35 out of 50 families (Fig. 3). Of these families, 14 showed supporting effects of temperature (positive mean annual temperature or negative elevation effects, $R^2$ = 0.14 and 0.06, respectively (all $P$ < 0.001)). Productivity negatively affected colour lightness in 18 families, and UVB radiation negatively affected colour lightness in 10 families ($R^2$ = 0.16 and 0.12, respectively; all $P$ < 0.001; for overall single linear regressions see Supplementary Fig. 3). Overall phylogenetic generalised least squares model testing for general relationships of raw colour lightness and the interaction of mean annual temperature, elevation, productivity and annual UVB showed insignificant and weak correlations (slope ± SE = 10.70 ± 5.18, t-value = 2.06, $P$ = 0.039, slope ± SE = 2.21 ± 1.68, t-value = 1.32, $P$ = 0.188, slope ± SE = −8.07 ± 1.28, t-value = −6.28, $P$ < 0.001, slope ± SE = −4.81 ± 2.85, t-value = −1.68, $P$ = 0.092, df = 3054, $R^2$ = 0.01, $\lambda$ = 0.46). Using the latitudinal distribution centre of each family (absolute values averaged across all species) showed that the relative importance of temperature and UVB from hierarchical partitioning analyses increased with increasing latitude, while the importance of productivity showed a U-shaped relationship with a flatter end at low latitudes (Supplementary Fig. 4).

Pathogen infections, particularly that of the chytrid fungus (*Batrachochytrium dendrobatidis*), are recognised as a major cause of population declines in amphibians[31]. Combining the uniquely comprehensive and detailed pathogen data available for chytridiomycosis infections with our data, showed that the severity of infections was generally higher for lighter-coloured species in regions with high productivity in 9 out of 20 families ($R^2$ = 0.41, Supplementary Fig. 5). Phylogenetic generalised least squares models testing for a general effect showed a consistent but less strong relationship of chytridiomycosis severity and the interaction of raw colour lightness with productivity (slope ± SE = 0.001 ± 0.0003, t-value = 4.21, $P$ < 0.001, $R^2$ = 0.01, $\lambda$ = 0.90).

Because frogs are mainly active at dusk or dawn and because thermoregulation is thought to provide developmental rather than activity benefits for nocturnal species, we analysed a subset of 820 African, European, and North American anurans with activity data. In line with previous findings for moths[34], the effects of all environmental factors, including proxies for temperature, were consistent among species with different activity patterns. Activity explained only a minor additional portion ($R^2$ = 0.03, $P$ < 0.001) of variance in colour lightness (Supplementary Table 1).

## Discussion

The origin and diversity of morphological variation in animals have fascinated biologists since the beginning of natural history research[35,36]. Frogs and toads show a remarkable spectrum of colours, and a rich body of research provides explanations for this phenomenon from the perspective of biotic interactions[10,37]. Many anuran species resemble vegetation and soil in colour to reduce predation risk[38]. Other taxa, such as the Dendrobatidae, have bright warning colours to signal toxicity to predators[10,39] and recent studies suggest that aposematic colourations drives diversification and genetic differentiation in anurans[37,40,41]. Our findings reveal three crucial functions of colour that are currently poorly understood for this taxon[15] and provide rigorous support for the general functional significance of colour lightness in ectotherms.

We document strong and globally consistent clines in colour lightness along gradients of temperature, productivity and UVB radiation that broadly expand our understanding of the physiology of

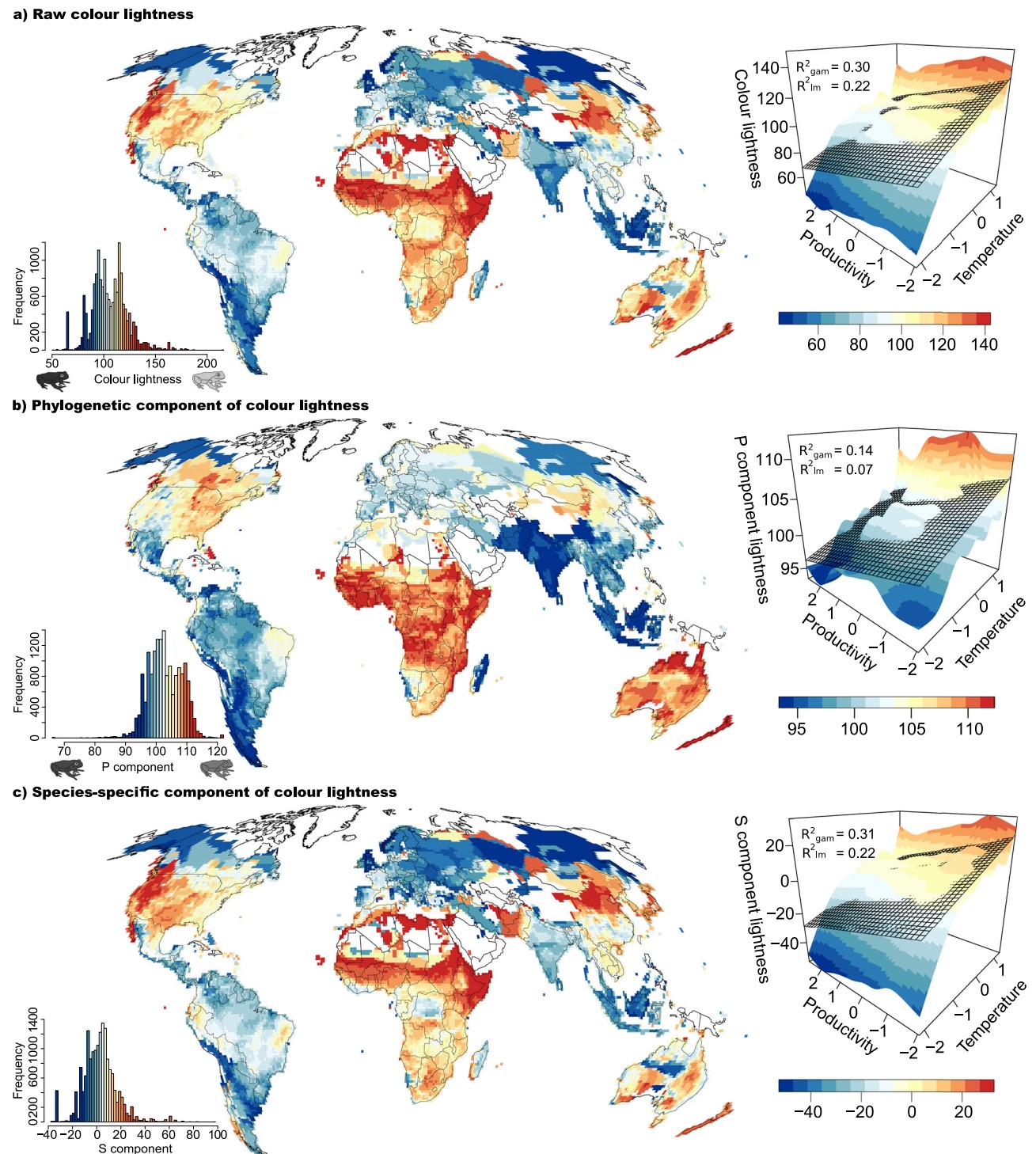

**Fig. 1 | Patterns in colour lightness variation.** Maps of **a** the raw colour lightness, **b** the phylogenetically predicted component of colour lightness, and **c** the species-specific deviation from this prediction across anuran assemblages ($n = 16,686$ assemblages; Mollweide projection). Plots on the right show spline-based smoothed and linear regressions (black grids) of the relationships of respective variables with z-scaled mean annual temperature and productivity (mean annual EVI). Values in the top-left corner indicate the explained variance from these regressions (raw linear model slope ± SE = 10.40 ± 0.16, $P < 10^{-16}$ (MAT), slope ± SE = −5.52 ± 0.15, $P < 10^{-16}$ (EVI); raw GAM, $F = 734.3$ (MAT), 151.8 (EVI); P component linear model slope ± SE = 1.80 ± 0.05, $P < 10^{-16}$ (MAT), slope ± SE = −0.99 ± 0.05, $P < 10^{-16}$ (EVI); P component GAM $F = 231.58$ (MAT), 92.69 (EVI); S component linear model slope ± SE = 8.59 ± 0.13, $P < 10^{-16}$ (MAT), slope ± SE = −4.54 ± 0.12, $P < 10^{-16}$ (EVI); S component GAM $F = 750.8$ (MAT), 160.7 (EVI). The significance of the correlation coefficients was tested using a two-sided $t$-test. The colour scale follows an equal-frequency classification. Copyrights for the frog icons are held by Ricarda Laumeier.

anurans. We show that anuran assemblages are generally darker coloured in regions with higher productivity and higher UVB radiation. Thus, corresponding to the predictions of Gloger's rule and the UVB protection hypothesis[11,15,22], the enhanced structural integrity, and greater UV absorbance of melanised cells seem to provide an advantage to darker assemblages in regions with high pathogen pressure and high UVB levels[11]. Both patterns are well supported for plumage colouration in birds[11], but thus far poorly understood in other taxa.

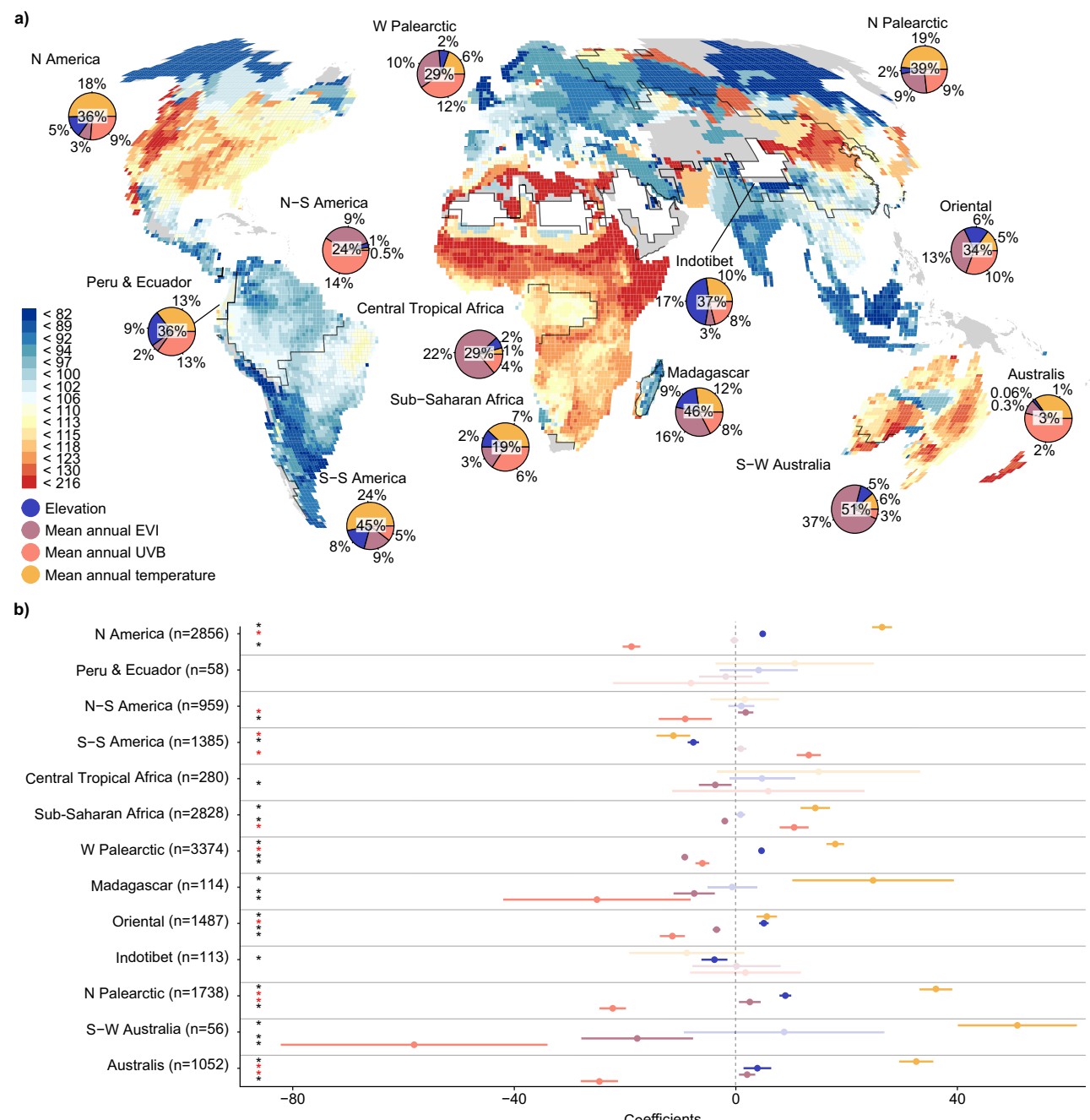

**Fig. 2 | Environmental predictors of colour lightness. a** Geographical pattern in the variation of raw colour lightness across anuran assemblages ($n = 16,300$ assemblages; Mollweide projection). The colour scale follows an equal-frequency classification. Pie charts inset in **a** show the independent contribution of environmental predictors (slices) as well as the overall $R^2$ (value in the circle centre) from hierarchical partitioning analyses per biogeographic realm. For areas in grey, no data were available. **b** Coefficient means and 95% confidence intervals of the interaction effects of environmental predictors (z-scaled) and biogeographical realm on the raw colour lightness of anuran assemblages (overall $R^2 = 0.52$). The significance of the correlation coefficients was tested using a two-sided $t$-test. Significant effects (no overlap with zero) are highlighted by an asterisk. Black asterisks indicate support for colour-based thermoregulation, pathogen resistance or UVB protection, whereas red asterisks indicate a converse trend. Only biogeographic realms including more than 50 assemblages are considered. Shaded colours show non-significant correlations.

Together with evidence for selected ectotherm taxa and regions[22,42], our findings suggest much broader implications of the protective functions of colour in animals. Particularly due to a geographical bias towards the North American and European faunas[11], the role of pathogen resistance has likely been underestimated in previous studies. Thus, we demonstrated that productivity is much more important for the colour lightness of anuran assemblages in tropical realms than in temperate realms and hence in realms where pathogen pressure is expected to be highest. However, to our knowledge, no prior studies have experimentally investigated the relationship between pathogen resistance and colour in amphibians.

Our results not only underline the significance of the protective functions of colour lightness but also reveal the central importance of colour lightness for thermoregulation in ectotherms. Size-based thermoregulation sensu Bergmann has dominated the discussion of ecogeographical patterns that underpin species' distributions. In

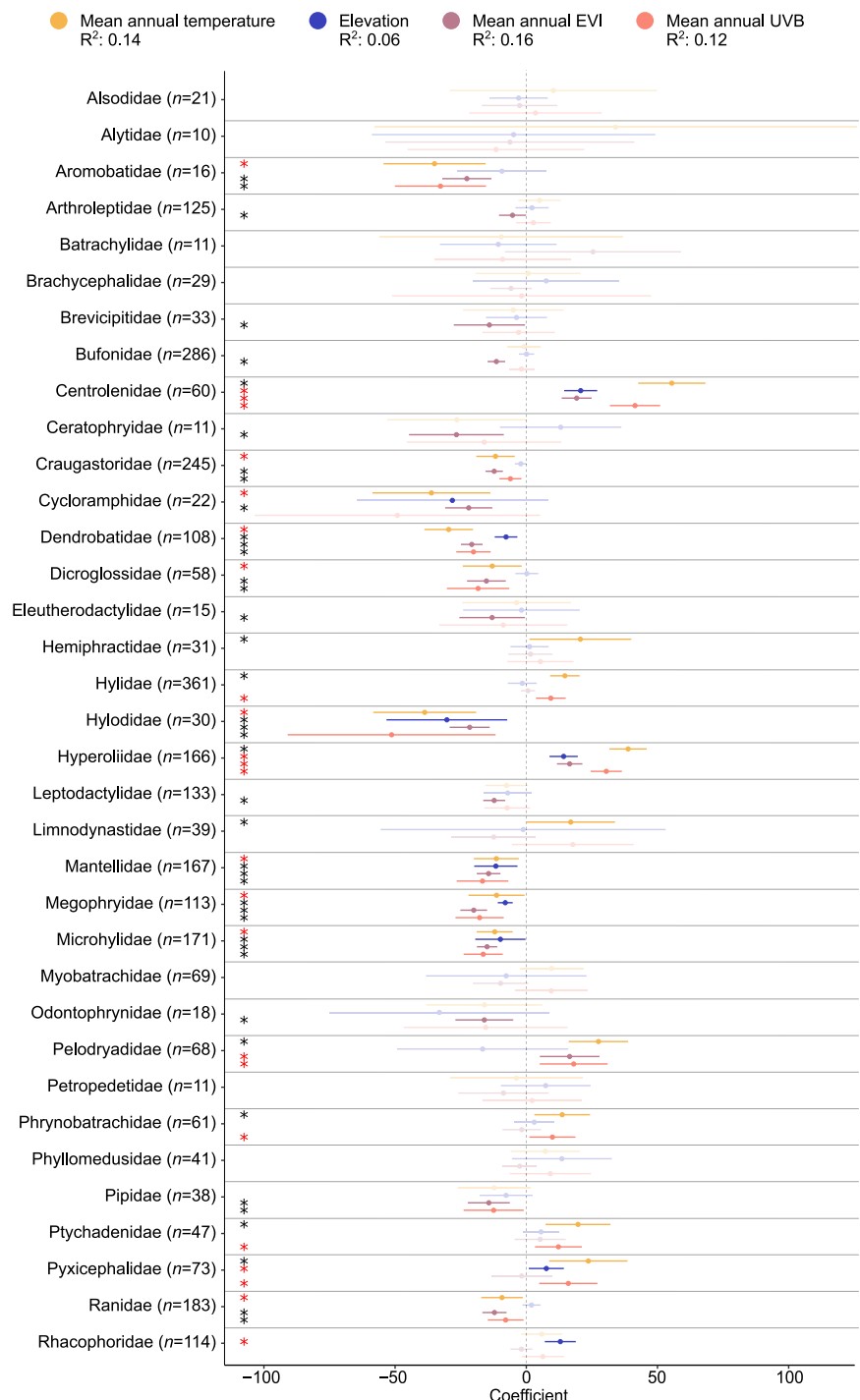

**Fig. 3 | Coefficient plot of single linear regression models of colour lightness and environmental variables at the species level (*n* = 2984 species).** Coloured bars indicate the 95% confidence interval, and dots indicate the mean standardised effect size of interactions of mean annual temperature, elevation, productivity (i.e., mean annual enhanced vegetation index, EVI) and mean annual UVB radiation for the families of anuran species. Bars that do not overlap with zero (dashed line) indicate significant effects. Black asterisks indicate support for colour-based thermoregulation, pathogen resistance or UVB protection, whereas red asterisks indicate a converse trend. Shaded colours show non-significant correlations. The significance of the correlation coefficients was tested using a two-sided *t*-test.

ectotherms, however, comprehensive studies on body size variation across species highlight that body size clines typically follow a converse pattern[43,44]. Our results suggest thermal melanism as an alternative and likely also additional mechanism, with high potential to resolve this weak and inconclusive support for size-based thermoregulation. Indeed, in the context of growing evidence for thermal melanism[13], colour lightness currently represents the most generally supported thermal adaptation in ectotherms. Our analysis of the

relative importance of the three investigated functions also uncovers a possible trade-off between colour-based thermoregulation and pathogen protection. Thus, a large part of the colour lightness variation at the warm end of the colour lightness–temperature relationship could be explained by pathogen resistance. Similarly, the species-level analysis highlights that families tend to either follow the expectation under thermal melanism or that from the protective functions of colour lightness (Fig. 3), whereas the signal of these functions is overall

weak and not general (PGLS: $R^2 = 0.01$, $\lambda = 0.46$). These patterns point to divergent selection pressures on certain functions associated with idiosyncrasies in the biogeographical history of regions and taxa (Supplementary Fig. 2).

Our general and strong support for colour lightness-based thermoregulation, pathogen resistance and UVB protection implies significant gains in the predictive power and accuracy of mechanistic and biophysical models[1,45] of ectothermic species when incorporating colour lightness. Moreover, accounting for the covariance of environmental drivers and colour lightness will help understand, forecast, and ultimately develop strategies to mitigate the biological consequences of environmental changes. Across European dragonflies, for instance, range shifts in accordance with colour-based thermoregulation changed the community composition towards lighter-coloured species on average[46]. Our results suggest that temperature is one but not the only driver of such changes and consequent threats to species. Multiple strong colour lightness–environment interactions and evolutionary constraints contribute to complexity in the spatial variation in colour lightness. On the one hand, darker anuran species in temperate regions are likely more vulnerable to extinction because they are already at the cold end of their tolerance limits (mountain tops or northernmost boundaries). On the other hand, lighter-coloured species may become more threatened because of their poorer resistance to pathogens. Temperature and UVB radiation were strong filters of the colour-lightness diversity of anurans (Supplementary Fig. 2) and of exceptionally high importance in temperate realms and families (Figs. 2 and 3). Thus, lineages with a predominantly temperate distribution might be susceptible to novel pathogens because they are incapable of overcoming phylogenetically conserved cold and UVB adaptations.

Leveraging uniquely extensive pathogen severity data for chytridiomycosis infections in amphibians—probably the most important pathogen threatening the biodiversity of this group—we show that lighter-coloured species in regions with a high productivity appear to be generally more infected. However, this effect is weak and not general across all considered species due to a strong dependence on the phylogenetic relationship of taxa (PGLS: $R^2 = 0.01$, $\lambda = 0.90$), but markedly differs among anuran families ($R^2 = 0.41$, Supplementary Fig. 5). These results provide phenomenological support for the impact of colour-based pathogen resistance (Gloger's rule), but it is important to note that experimental evidence for colour-based pathogen resistance is lacking in amphibians. Thus, while in depth analysis of such trait-environment relationships and the threats of species are beyond the scope of our study, our findings motivate further research along this avenue. Together these results suggest that multiple, partly competing functions of colour lightness can result in the loss of a distinct part of the overall functional and phylogenetic diversity (Supplementary Fig. 2). Since threats to biodiversity, including pathogens and climate change, interact[47], we also urge researchers to account for the multifunctional nature of traits when making mechanistic predictions of species' responses.

## Methods
### Colour lightness estimation
The colour lightness of 3386 anuran species was estimated based on photos and illustrations from 27 amphibian field guides and monographs (for sources and details, see Supplementary Table 2). These literature sources were selected because they cover the entire known amphibian fauna of the respective regions. Due to missing distribution maps for several species, our final dataset covered 3059 species (4305 individuals; 1.4 individuals on average, see Supplementary Table 3). We focused on anurans instead of all amphibian species because the two remaining orders of amphibians differ from frogs and toads in their body shape, distribution, and ecology. Specifically, Caudata (9%) are mostly absent from tropical regions and comprise several semi-

fossorial and aquatic species. Gymnophiona (3%) are exclusively distributed in the tropics and are mainly fossorial and aquatic. In contrast, except for Antarctica, anurans occur on all continents and mainly comprise semi-aquatic, terrestrial and arboreal taxa.

To estimate the colour lightness of the dorsal surface of individuals (i.e., the part of the body that is exposed to sunlight) from images, we used a standardised colour wheel with 72 colours (Supplementary Fig. 6). The wheel comprises 18 rays, each consisting of four colours with different saturation levels (S: 10, 40, 70, 100%). In addition to assigning a colour per individual, we also visually estimated the coverage of the ventral surface colouration in 5% steps (5, 10,…, 100). To obtain a proxy for melanisation, we calculated the coverage-weighted mean of the red (R), green (G), and blue (B) colour channels of the colours of individuals (Supplementary Fig. 6, Supplementary Table 4). This average colour lightness value can range from 0 (black) to 255 (white).

$$\text{Colour lightness} = \text{coverage1} * \left(\frac{R_1 + G_1 + B_1}{3}\right) + \text{coverage2} * \left(\frac{R_2 + G_2 + B_2}{3}\right)$$
(1)

We did not distinguish between sexes, as they were typically not indicated. Instead, colour lightness values of all available individuals within species were averaged (morphs and sexes). Although it is known that anurans change colour to regulate their body temperature (e.g., phenotypic plasticity)[48,49], we have not been able to take this into account when collecting colour data because of limited information in field guides. Note that previous studies on butterflies (assessed in the visual spectrum based on illustrations or photos) and yeast have demonstrated a strong correlation between colour estimates and species' body temperature[46] ($r = -0.76$) and the temperature access rate of yeast[50]. This suggests that colour lightness assessed in the visible spectrum is a suitable surrogate for overall reflectance. We herein chose a visual assessment of colour lightness instead of a digital image analysis to facilitate the exclusion of reflections (wet skin) and shaded areas. Another advantage of visual estimation is that it allows the integration of multiple available sources of in situ images despite differences in the camera–object angle and light conditions (e.g., due to the use of flash). This is particularly important for amphibians, as they fade in colour after their death, impeding the use of the only comparatively rich source of images—material from museum collections—for colour assessments. The main colour assessment was performed by a single person (Ricarda Laumeier). However, a subset of species was also assessed by a second person (Antje Schmidt), and analyses of these data showed agreement between the assessments ($R^2 = 0.65$, slope $\pm$ SE $= 0.60 \pm 0.04$, intercept $= 24.45$, $P < 0.001$, $n = 107$ species; Supplementary Fig. 7). We also acknowledge the systematic observer bias documented by this comparison, which urges the need to align estimates based on species overlap when data assessed by different observers should be integrated. In addition, we compared estimates obtained with the colour wheel approach with estimates calculated from RGB values of manually probed pixels of distinct colours based on scans for 107 species (scanner: MUSTEK A3 2400 S)[51]. For both sets of colours the same coverage weight was used. Although the scans were generally redder than the original illustrations and differed in light conditions, the measurements of the two approaches were similar ($R^2 = 0.65$, slope $\pm$ SE $= 0.86 \pm 0.06$, intercept: 28.00, $P < 0.001$, $n = 107$ species; Supplementary Fig. 7).

### Distribution data
Vector distribution maps[52] for anuran species were reassigned to an equal-area grid (MGRS, cell size of approximately 100 km × 100 km) with functions provided in the R package sf [53]. Grid cells that contained more than 50% water were excluded and this criterion only removed two species of our final dataset. Distribution data were

transformed and mapped to a Mollweide equal-area projection with functions of the R package sp[54].

### Environmental data

Based on predictions of the thermal melanism hypothesis (colour-based thermoregulation), Gloger's rule and the UVB protection hypothesis, four environmental proxies for temperature (mean annual temperature (v2[55]), elevation[56]), productivity (mean annual EVI[57]), and UVB radiation[58] were derived from global high-resolution data. Because previous studies suggested that camouflage can result in a colour lightness pattern similar to that predicted for colour-based pathogen resistance, with species being darker in regions with a higher tree cover[11], we also retrieved land cover data for all tree categories at a resolution of 1 km[59] and subsequently summed up their coverage. Values for each of these variables were averaged across species' ranges for species-level analyses and across grid cells of our equal-area grid for assemblage-level analyses. As our supplementary analysis confirmed that productivity drives tree cover and that tree cover does not affect the colour lightness of assemblages, we did not include tree cover in our main analyses (Supplementary Table 5).

### Phylogenetic autocorrelation

To construct a comprehensive phylogeny for our subset of species, 230 species were added to the currently most complete molecular information[60] at the respective genus level, and the intra-genus relationships were randomly resolved with functions of the R package phytools[61]. Subsequently, species without colour lightness data were pruned from this tree for phylogenetic comparative analyses. We tested for a phylogenetic signal (Pagel's lambda[62]) in the colour lightness variation of anuran species using functions of the R package phytools[61] (for an ancestral reconstruction of colour variation see Supplementary Fig. 8). To evaluate the phylogenetic uncertainty introduced by the inclusion of species that were added to the original phylogeny, we also calculated the phylogenetic signal based on the original phylogeny only. Because the strength of the phylogenetic signal in colour lightness was similar between the original tree ($\lambda = 0.51$) and 10 alternatives of the extended tree [median $\lambda = 0.50$], one phylogeny from the set of extended trees was randomly selected for subsequent analyses. Finally, the phylogenetic signal for colour lightness and environmental aggregates was calculated based on this tree to test for phylogenetic niche conservatism. Because of a strong impact of the phylogenetic relationship of species, colour lightness was decomposed into its phylogenetically predicted part and the deviation from this prediction based on the most recent phylogenetic tree for amphibians[60] using Lynch's comparative method[33]. To perform Lynch's comparative method, we used the package ape[63]. The advantage of this method, compared to merely accounting for phylogenetic autocorrelation, is that it allows separate analyses of the long-term evolutionary (P-component) and short-term (species-specific, S-component) significance of trait–environment relationships. For assemblage-level analysis, these components were averaged across co-occurring species. To test for the generality of trait-environment relationships and the impact of trait-environment relationships on chytridiomycosis severity, we accounted for phylogenetic autocorrelation in our data by using phylogenetic generalised least square models[64].

### Spatial autocorrelation

Macroecological patterns typically show spatial autocorrelation (i.e., nearby locations have more similar values than those farther apart), which reduces the effective sample size and thereby leads to false model estimates[65]. To account for spatial autocorrelation in our models, we repeated assemblage-level analyses with GAM of colour lightness and environmental factors that included a smoothed (trend surface) term of the geographical coordinates of each assemblage using functions of the R package mgcv[66]. Spatial correlogram (R package ncf[67]) based on the residuals of linear models and these GAMs show that in the latter spatial autocorrelation was reduced to a minimum (Supplementary Fig. 9).

### Regression models

Residuals of all models were checked, and if necessary, predictors were transformed to meet the assumption of normality. All predictors were scaled and centred (z-scaled). All analyses were performed with the statistical software R[68]. Trait–environment relationships were studied at the species and assemblage levels to examine differences in the importance of thermoregulation, UV protection and pathogen resistance between clades and regions in taxonomically and spatially specific contexts, respectively.

**Assemblage-level analyses.** In assemblage-level analyses, we fitted multiple regressions for the relationships of environmental predictors with raw colour lightness, its phylogenetically predicted component, and its species-specific component as well as colour lightness diversity. Assemblages for which colour lightness data were available for less than 33% of species were removed to reduce the effects of incomplete representation (Supplementary Fig. 10), leaving 16,686 of the initial 17,170 assemblages. Colour lightness diversity was calculated as the standardised effect sizes of mean pairwise distances of co-occurring species minus a null model of 1000 alternative species sets with the same number of species randomly drawn from the species pool of the respective biogeographical realm[69]. This measure quantifies colour lightness diversity as under-dispersed, over-dispersed, or randomly dispersed for assemblages with more than one species ($n = 14,715$ assemblages). At our coarse grain, trait under-dispersion (negative values) can be interpreted as the filtering of species by a dominant driver (in our case, environmental factors associated with the functions of colour), whereas trait over-dispersion may indicate the presence of multiple concurrent drivers[70]. At a fine spatial grain where species are more likely to interact with one another and for traits that are associated with biotic interactions such as resource competition or trophic position, over-dispersion is commonly interpreted as trait-mediated competition. For the main assemblage-level analyses, we fitted both multiple linear regressions and a GAM that included a trend surface term to account for spatial autocorrelation. In addition, we assessed regional differences in the relative importance of presumed environmental drivers of the colour lightness and colour lightness diversity of anuran assemblages by repeating our main assemblage-level analysis for the biogeographical realms of amphibians, as identified by Holt et al.[69]. For these analyses, only assemblages of biogeographical realms with more than 50 assemblages were considered, leaving 13 of 19 realms.

**Species-level analyses.** To understand differences in the importance of different environmental drivers among major lineages, we repeated the analysis of raw colour lightness, including interactions between all pairs of the environmental predictors and anuran families. In these models, species belonging to families with fewer than 10 species were excluded, which reduced the dataset to 2984 species (Fig. 3). In addition, for each family, the relative contribution of environmental drivers of colour lightness from hierarchical partitioning analysis was plotted against the average absolute latitude of species to assess latitudinal changes in the dominant driver (Supplementary Fig. 4). To test for an effect of activity, we fitted a multiple linear regression of colour lightness with the interaction terms of each environmental variable and activity time (diurnal, nocturnal or diurnal and nocturnal) (Supplementary Table 1) for 820 European, North American and African anuran species from the respective literature sources[71,72].

To exemplify links between colour lightness, productivity (pathogen pressure), and infection rate, we combined our data with experimental data and expert knowledge on the severity of chytridiomycosis fungi infection for 258 species taken from Scheele et al.[31] and conducted a single regression of the interaction between colour lightness and productivity (Annual EVI) on chytridiomycosis severity for each family. As a control (species unaffected), we considered all species that co-occurred with the species affected by chytridiomycosis. Our total data set includes 1291 species, 258 from Scheele et al.[33] which matched with our dataset and 1033 species that were classified as unaffected because they occurred in the same grid cell as the affected species. All families with fewer than 9 species were excluded (Supplementary Fig. 5).

### Reporting summary

Further information on research design is available in the Nature Portfolio Reporting Summary linked to this article.

## Data availability

The assemblage data generated in this study have been deposited in the figshare database under accession code https://doi.org/10.6084/m9.figshare.22303480. Raw climate data were obtained from CHELSA[55], accessible at https://chelsa-climate.org/. Elevation data[56], Enhanced Vegetation Index (EVI)[57] data, and land-cover data[59] can be downloaded from https://www.earthenv.org. Global UV-B radiation data[58] can be accessed via https://www.ufz.de/gluv/index.php?en=32435. The amphibian phylogeny[60] can be found at https://vertlife.org/data/amphibians/. Biogeographic realms are available at https://macroecology.ku.dk/resources/wallace.

## Code availability

No custom code or mathematical algorithm has been used.

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

## Acknowledgements

We acknowledge support in assessing colour lightness and data validation by Antje Schmidt. S.P. acknowledges support from the Alexander von Humboldt Foundation.

## Author contributions

R.L. compiled the data and led the analyses. S.P. and R.B. conceived the study. R.L. wrote the first draft. S.P. assisted in writing, data processing and data analysis. M.-O.R. provided additional data. R.L., M.B., M.-O.R., S.B., R.B. and S.P. contributed substantially to subsequent revisions. All authors read and approved the final paper.

## Funding

## Competing interests

The authors declare no competing interests.
