## [Peer Review File · Nature Communications]

The global importance and interplay of colour-based protective and thermoregulatory functions in frogsREVIEWER COMMENTS

Reviewer #1 (Remarks to the Author):

In this manuscript the authors use a large-scale database of anuran colour to test whether existing ecogeographical rules of colour (thermal melanism hypothesis, Gloger's rule, and also the effects of UV radiation) can explain colour variation between species and frog assemblages at a global scale.

In general, the find that predictions from both ecogeographical rules are supported, revealing the pervasive effects that climatic variation has had on the evolution of anuran colours. These data add to the growing body of evidence documenting climatic effects on colour across the tree of life. I therefore find that this is a highly timely study that helps us to broaden the taxonomic scope of such climatic effects.

I do however think that in its current state the manuscript does not reach its full potential and there are several points that require improvement. Below I first highlight the major points that I think need addressing and then detail specific comments.

Major suggestions

1. The introduction gives the impression that the body of literature dealing with links between colour and climate is rather small. This is not the case and there have been multiple papers across a variety of taxa both ecto- and endotherms describing such associations. As a matter of fact, the current paper is timely precisely because of this revival of the ecogeographical patterns of colour field! I suggest that the authors include some of this literature to provide a more balanced background.
2. A little bit more introduction on the mechanisms behind amphibian integumentary coloration would be good here. When discussing frog colour in terms of thermoregulation it may be good to keep some older work in mind as well (eg (Schwalm et al., 1977)).
3. It might be good to explore simple patterns of climatic variation (e.g. annual mean temperature and annual precipitation) to ease comparison with other studies. Both ecogeographical rules tested (thermal melanism hypothesis and Gloger's rule) have been formulated in terms of these simple climatic variables.
4. Scatterplots depicting the correlation between raw species-level data and predictors would be a nice addition to convey patterns to the reader. These are more important than the realm-specific analyses presented in Figs 2-3. These could be added to the suppl. material.
5. I would suggest that the main analysis should be a species-level analysis since this is closest to the level of adaptation. Assemblage-level analyses are attractive mainly because they nicely illustrate variation on the map, but they are in itself only an emergent pattern.
6. The analytical approach is very hard to follow. In the results section it is hard to separate analyses at the assemblage-level from analyses at the species level. In the methods, more information is needed to be able to repeat the analyses. Please indicate how you implemented Lynch's method and what software was used. How did you obtain R² values? Are these marginal R² or conditional R² values? Would results be different if you would have used a different approach (e.g. PGLS)?
7. I am not convinced that partitioning analyses across realms should be given so much prominence in the main manuscript. While the results are important I would suggest that the figures would be better placed in the suppl. material. This way you can free some space to depict patterns at the species level better.
8. In some places, I feel that the authors jump to conclusions too quickly, this is particularly the case when linking pathogen resistance to colour. If this interpretation has good support it would be

good to explain it better, citing the appropriate references (e.g. is there anything known about links between susceptibility to chytrid fungus and colour in amphibians?), if not make it clear that these are hypotheses that require further testing.

Specific comments (Ln indicates line number)

Ln 23-25 This part may be selling the growing body of knowledge in this field a bit short. There have been many studies (some quite large scale) uncovering the links between coloration and climatic factors in ecto- and endotherms. A few examples: (Smith et al., 2016; Medina et al., 2018; Wisocki et al., 2019; Krah et al., 2019; Marcondes & Brumfield, 2019; Munro et al., 2019; Nicolai et al., 2020; Delhey et al., 2021; Goldenberg et al., 2021; Kang et al., 2021; Idec et al., 2023).

Ln 58-60 there are other mechanisms associated with Gloger's rule, prominent among them is camouflage. Perhaps it would be good to mention these alternatives as well.

Ln 87 I suspect the true r^2 value associated to the biological effects is lower, often these high r^2 values are mainly driven by the procedures used to account and control for spatial autocorrelation. Is there a possibility to separate them? How did you compute them? This also applies to lines 103-105.

Ln 90 it would it be good to separate better assemblage-level analyses from species-level analyses. My preference would also be to first present the species-level analyses and then the assemblage-level analyses. As the latter is simply an emergent pattern from species-level data and distributions.

Ln 109-118 From reading this section it is unclear whether these are analyses done at the species- or assemblage-levels.

Ln 147 this is confusing, did you not refer to species-level data before (phylogenetic signal estimates require species-level data [how were these computed?])

Ln 187-188 in birds the selection forces behind Gloger's rule are unclear, but camouflage may be playing an important role as well. I think that it would be good to expand on the potential mechanistic links between pathogen resistance and melanin pigmentation in anurans. In birds the mechanism is mainly concerned with plumage resistance to keratinolytic bacteria something that does not apply to anurans.

Ln 199-200 revise writing.

Ln 209-211 again, you are presenting pathogen-resistance as a fact, when in reality it is more of an hypothesis.

Ln 219-221 I like this sentence!

Fig 2 could be included in the suppl. material.

Ln 443-446 does this method yield similar conclusions to those obtained through the use of PGLS models? I think that it would be good to test this, since PGLS analyses are by far more common in the literature, and this would allow more targeted comparisons between taxa.

Ln 449-450 how did you assess whether this approach effectively accounted for spatial autocorrelation? Plots, Moran's I , etc.?

Ln 477-479 I feel that you are too assertive here, the data you generated does not allow you to say this, it simply indicates whether variation in colour lightness is higher or lower than expected. Whether this is caused by a dominant driver or multiple drivers or something else is not something that you can determine with these data. While you can speculate that this is the case in the discussion here it sounds to clear-cut.

Extended Data Fig. 1 what is the reference for the colour wheel used?

Extended Data Fig. 4 This figure is very hard to follow, I think these data should be presented as a table.

References

Delhey, K., Dale, J., Valcu, M. & Kempenaers, B. 2021. Migratory birds are lighter coloured. *Curr. Biol.* 31: R1511–R1512.

Goldenberg, J., D’Alba, L., Bisschop, K., Vanthournout, B. & Shawkey, M. 2021. Substrate thermal properties influence ventral brightness evolution in ectotherms. *Commun. Biol.* 1–10. Springer US.

Idec, J.H., Bishop, T.R. & Fisher, B.L. 2023. Using computer vision to understand the global biogeography of ant color. *Ecography (Cop.)*. 2023(3): e06279.

Kang, C., Im, S., Young, W., Yunji, L., Fox, D.S.- & Huertas, B. 2021. Climate predicts both visible and near- infrared reflectance in butterflies. 1869–1879.

Krah, F., Büntgen, U., Schaefer, H., Müller, J., Andrew, C., Boddy, L., et al. 2019. European mushroom assemblages are darker in cold climates. *Nat. Commun.* 10: 2890. Springer US.

Marcondes, R.S. & Brumfield, R.T. 2019. Fifty shades of brown: Macroevolution of plumage brightness in the Furnariida, a large clade of drab Neotropical passerines. *Evolution (N. Y.)*. 73: 704–719.

Medina, I., Newton, E., Kearney, M.R., Mulder, R.A., Porter, W.P. & Stuart-Fox, D. 2018. Reflection of near-infrared light confers thermal protection in birds. *Nat. Commun.* 9: 3610. Springer US.

Munro, J.T., Medina, I., Walker, K., Moussalli, A., Kearney, M.R., Dyer, A.G., et al. 2019. Climate is a strong predictor of near- infrared reflectance but a poor predictor of colour in butterflies.

Nicolai, M.P., Shawkey, M.D., Porchetta, S., Claus, R. & D’Alba, L. 2020. Exposure to UV radiance predicts repeated evolution of concealed black skin in birds. *Nat. Commun.* 11: 2414.

Schwalm, P.A., Starrett, P.H. & McDiarmid, R.W. 1977. Infrared reflectance in leaf-sitting neotropical frogs. *Science (80-)*. 196: 1225–1226.

Smith, K.R., Cadena, V., Endler, J.A., Porter, W.P., Kearney, M.R. & Stuart-Fox, D. 2016. Colour change on different body regions provides thermal and signalling advantages in bearded dragon lizards. *Proc. R. Soc. B Biol. Sci.* 283: 1–9.

Wisocki, P.A., Kennelly, P., Rivera, I.R., Cassey, P., Burkey, M.L. & Hanley, D. 2019. The global distribution of avian eggshell colours suggest a thermoregulatory benefit of darker pigmentation. *Nat. Ecol. Evol.* doi:10.1038/s41559-019-1003-2.

Reviewer #2 (Remarks to the Author):

I have now read with interest the manuscript titled ‘Frogs reveal the global importance and interplay of colour-based protective and thermoregulatory functions’. Laumeier et al. set out to understand the drivers of assemblage-level colour lightness in frogs using data from 40% of all species. Specifically, they investigated three main drivers of colour lightness – thermoregulation, pathogen resistance, and UVB protection – using appropriate environmental proxies. Laumeier et al. show color lightness to decrease in colder regions and regions with higher pathogen pressure and UVB radiation.

I enjoyed reading the paper, and I think it’s a topic that will be of broad interest. However, I cannot recommend the manuscript for publication in its current form. I will explain this in detail below.

#####

First, the study quantified colour lightness using a standardized colour wheel. This was achieved by visually matching photographs from field guides by a single author (L.B.). While visual estimation by human observers has its advantages (easily quantifiable for a large number of species), this method does not capture variation in the other spectrum – ultraviolet or infrared (which together account for >50% of solar radiation). The dataset is thus based on human perception of colours (visible range).

Recent studies have shown infrared reflectance may also play a critical role in heat gain (e.g., Medina et al. 2018 and also Schwalm et al. 1977). I would like the authors to address this issue by running separate analyses by using data from actual reflectance values when available from the literature. For example, see Clusella-Trullas et al., 2008 who collected reflectance data from literature sources to test the thermal-melanism hypothesis in lizards. I think a similar analysis can be conducted for a few species to support the main claims in this study or at least show color lightness measured from different methods is tightly related. In any case, the authors must better explain and provide justification for using human-based classifications (only visible spectrum) and make readers aware of the limitation.

Schwalm, P.A., Starrett, P.H. and McDiarmid, R.W., 1977. Infrared reflectance in leaf-sitting neotropical frogs. *Science*, 196(4295), pp.1225-1226.

Medina, I., Newton, E., Kearney, M.R., Mulder, R.A., Porter, W.P. and Stuart-Fox, D., 2018. Reflection of near-infrared light confers thermal protection in birds. *Nature communications*, 9(1), p.3610.

Clusella-Trullas, S., Terblanche, J.S., Blackburn, T.M. and Chown, S.L., 2008. Testing the thermal melanism hypothesis: a macrophysiological approach. *Functional ecology*, pp.232-238.

Second, I thank the authors for showing interobserver differences in color lightness classification (Extended data Fig.2). However, there is only a moderate correlation between the two observers ($R^2 = 0.65$), suggesting there is still subjective bias in rating color lightness. To make the argument much stronger, the authors can directly measure color lightness from the scanned photographs (pixel level) for a few species and compare them with lightness classified by human observers. This can be done by extracting pixel colour value manually from images using software such as ImageJ (see Roberts et al. 2022 for a similar approach). I suggest authors show this objective measure of color lightness results in the supplement.

Roberts, S.M., Stuart-Fox, D. and Medina, I., 2022. The evolution of conspicuousness in frogs: When to signal toxicity?. *Journal of Evolutionary Biology*, 35(11), pp.1455-1464.

Third, the statistical analysis must be improved. It is unclear how authors accounted for multicollinearity in their analysis. I could not find details in the methods section. It is well known that temperature and UV radiation are tightly correlated with elevation, and productivity is related to temperature. Consequently, the effect sizes and direction of regression co-efficient presented in the ms are not reliable. Therefore, any inference made on the contribution of each variable in explaining lightness value (for e.g., Lines 28-30) is potentially incorrect.

Further, these environmental predictors may or may not act in synergy (see Delhey 2019 review) – which was not addressed in the current manuscript. To test if these predictors are acting independently or in synergy the author could use a Structural Equation Modeling analysis and see if the path of influence between predictors and response variable is independent or not. There are R packages available for SEM methods ('lavaan' or 'phylopath'). I strongly suggest authors present their results in a path-analysis framework.

Fourth, another major issue with the ms is attributing the effect EVI on lightness to pathogen pressure alone. Productive environments are likely associated with closed habitats (i.e., habitats with low light levels, such as high canopy trees especially in the tropics), where animals are predicted to be darker in coloration, potentially due to crypsis or other reasons. Therefore, the fact that authors find a strong effect of EVI on lightness could be due to the selection for signaling based on light conditions (in addition to pathogen?). See the below study by Marcondes et al. 2021 who showed forested birds have darker plumage even after controlling for the climate. See also Delhey et al. 2019 who used the percentage of tree cover as a predictor.

I strongly suggest authors include a proxy for the light environment (tree cover or forested habitat type) as a predictor (see below papers). As suggested above, incorporating this in a path analysis framework will enable authors to tease apart the main driver of lightness in the tropics.

Marcondes, R.S., Nations, J.A., Seeholzer, G.F. and Brumfield, R.T., 2021. Rethinking Gloger's rule: climate, light environments, and color in a large family of tropical birds (Furnariidae). *The American Naturalist*, 197(5), pp.592-606.

Delhey, K., Dale, J., Valcu, M. and Kempnaers, B., 2019. Reconciling ecogeographical rules: rainfall and temperature predict global colour variation in the largest bird radiation. *Ecology Letters*, 22(4), pp.726-736.

Finally, I found the results section disconnected from the introduction. For instance, the authors directly present the results from regression analysis where environmental predictors are used as a proxy to test the three main drivers of lightness. A reader may find it hard to connect these results with the main mechanisms in the introduction without reading the methods. I suggest authors include a brief aim and the method employed to address them at the start of each results section (1 or 2 sentences).

I also find that many key methodological details are not described in the methods section (see specific comments below). For example, no justification was provided on why specific environmental variables were chosen and the exact mechanism linking that variable with the three main drivers. I understand the journal restrictions on the word limit. However, I suggest authors include detailed methods with all the necessary details as done in a normal full-length paper in the supplement.

#####

Specific comments

1.Line 56: Also known as 'Bogert's rule'.

2.Line 58: The below study might be highly relevant.

Fu, T.T., Sun, Y.B., Gao, W., Long, C.B., Yang, C.H., Yang, X.W., Zhang, Y., Lan, X.Q., Huang, S., Jin, J.Q. and Murphy, R.W., 2022. The highest-elevation frog provides insights into mechanisms and evolution of defenses against high UV radiation. *Proceedings of the National Academy of Sciences*, 119(46), p.e2212406119.

3.Line 59: 'darker species tend to be more resistant to pathogen' – the mechanism should be added here. Gloger's rule was proposed for endotherms, and it has not been shown in ectotherms (at least in amphibians and reptiles). Consider adding citations for ectotherms backed up by clear mechanistic explanations.

4.Line 60: Not only parasites but also for signalling (e.g., crypsis). See my general point #4.

5.Line 68: The authors did not directly test any of the three drivers. Consider stating that you tested these mechanisms with environmental proxies.

6.Line 84: 'increasing temperature...and decreasing productivity' this could simply be due to multicollinearity. See general comment #3. This also applies to realm-level analysis that highlights the importance of each predictor.

7.Line 110: 'generally weaker but remained robust'. It is unclear whether if authors standardized their regression coefficients to make such a comparison across the methods. Clarify and add details in the methods section.

8.Line 176: But according to recent studies on birds (general comment #4), the influence of signaling strategies on colour evolution cannot be ignored. See the below review.

Rojas, B., 2017. Behavioural, ecological, and evolutionary aspects of diversity in frog colour patterns. *Biological Reviews*, 92(2), pp.1059-1080.

9.Line 185: I am not entirely convinced by the structural integrity idea in Anurans. The cited examples are for birds and ants, which probably make sense due to external structures (feathers and exoskeletons).

10.Line 376: The 2021 version of IUCN has data for about ~6700 species, I always assumed color lightness values were hard to obtain relative to distribution. I suggest authors provide family-level data coverage for both in the supplement.

11.Line 385: Are the images photographs of actual animals or illustrations? If from illustrations, the colours may not be accurate (drawn by artists). Is it possible to show that there is no difference between the two sources?

12.Line 389: 5% as in saturation levels?

13.Line 389-390: Consider proving the formula for the weighted mean. Also mentioned this index as color-lightness to avoid confusion (same as what authors refer to as melanization?)

14.Line 393: Consider proving the total number of images analyzed and the average number of images per species.

15.Line 412: I am not sure on what basis the authors used a 50% cut-off – consider adding relevant references or running sensitivity analysis for other cut-off values.

16.Line 412: Regarding the above point, it is unclear how the authors treated narrow-ranged

species (1-2 grid cells). Most of these species' ranges do not cover the entire grid cell (usually, endemics have <50% coverage). Did the authors remove all such species from the analysis? Please clarify.

17.Line 448: I could not confidently conclude that there is no spatial autocorrelation without a test on residuals (Moran's I?). Please provide test results in the supplement.

18.Line 493: Should this be the proportion of individuals per family rather than absolute numbers?

19.Line 499 to 500: It is unclear from where the authors obtained the activity data from. Consider citing the source reference.

20.Line 1: Minor point. The title of the paper can be changed to 'The global importance and interplay of colour-based protective and thermoregulatory functions in frogs'. I also suggest replacing the mechanisms with actual predictors (e.g., 'thermoregulatory' with 'temperature') in the title, as done in Delhey et al. 2019.

REVIEWER COMMENTS

Reviewer #1 (Remarks to the Author):

General comment: In this manuscript the authors use a large-scale database of anuran colour to test whether existing ecogeographical rules of colour (thermal melanism hypothesis, Gloger's rule, and also the effects of UV radiation) can explain colour variation between species and frog assemblages at a global scale.

In general, the find that predictions from both ecogeographical rules are supported, revealing the pervasive effects that climatic variation has had on the evolution of anuran colours. These data add to the growing body of evidence documenting climatic effects on colour across the tree of life. I therefore find that this is a highly timely study that helps us to broaden the taxonomic scope of such climatic effects.

I do however think that in its current state the manuscript does not reach its full potential and there are several points that require improvement. Below I first highlight the major points that I think need addressing and then detail specific comments.

General response: Thank you for this positive and constructive feedback. We have carefully considered all your comments and made revisions accordingly. The supplementary analyses and references you suggested underlined the robustness of the results and strengthened our conclusions. We are very grateful for the time you have invested to improve our manuscript.

Note that the line references in comments are from the original document, whereas the line references in our responses refers to the revised version of the manuscript.

Comment: Major suggestions

1. The introduction gives the impression that the body of literature dealing with links between colour and climate is rather small. This is not the case and there have been multiple papers across a variety of taxa both ecto- and endotherms describing such associations. As a matter of fact, the current paper is timely precisely because of this revival of the ecogeographical patterns of colour field! I suggest that the authors include some of this literature to provide a more balanced background.

Response: We now reference a broader array of literature on the relationship of colour and climate.

Comment: 2. A little bit more introduction on the mechanisms behind amphibian integumentary coloration would be good here. When discussing frog colour in terms of thermoregulation it may be good to keep some older work in mind as well (eg (Schwalm et al., 1977)).

Response: We revised our introduction, included additional citations and provided a more detailed background.

Comment: 3. It might be good to explore simple patterns of climatic variation (e.g. annual mean temperature and annual precipitation) to ease comparison with other studies. Both ecogeographical rules tested (thermal melanism hypothesis and Gloger's rule) have been formulated in terms of these simple climatic variables.

Response: In the supplement (Supplementary Figure 1), we now provide linear regressions of assemblage level colour lightness and environmental variables related to thermoregulation [Mean Annual Temperature (MAT) and Elevation], pathogen resistance [i.e. Gloger's Rule; Annual Precipitation (AP), Annual Enhanced Vegetation Index (EVI), the interaction of MAT and AP] and UV protection [Annual UVB radiation].

Comment: 4. Scatterplots depicting the correlation between raw species-level data and predictors would be a nice addition to convey patterns to the reader. These are more important than the realm-specific analyses presented in Figs 2-3. These could be added to the suppl. material.

Response: We now added scatterplots for raw species-level data and predictors to the supplement (Supplementary Fig. 2).

Comment: 5. I would suggest that the main analysis should be a species-level analysis since this is closest to the level of adaptation. Assemblage-level analyses are attractive mainly because they nicely illustrate variation on the map, but they are in itself only an emergent pattern.

Response: We now focus more on the species-level results in our study, but several important insights cannot be retrieved at this level and hence a greater species-level focus would severely change the scope of our article. For instance, biogeographical patterns, differences in the relative importance of drivers among the biogeographical realms and the colour lightness diversity are purely spatial. In addition, the data was quantified from literature sources with a geographical scope, so that the strength of our data is the high spatial coverage rather than the species coverage. The species-level approach implies a reduction of the spatial detail for the sake of trait detail. If an environmental predictor is reduced to a single value (e.g. average elevation across the range) any information about its location is lost and in some cases the central values reflect *de facto* unsuitable areas (a species occurring in the Scandinavian alps (at 500-900m) and the alps (1100-1300m)) can technically result in an estimated average elevation between 900-1100m although it has not been recorded there.

Comment: 6. The analytical approach is very hard to follow. In the results section it is hard to separate analyses at the assemblage-level from analyses at the species level. In the methods, more information is needed to be able to repeat the analyses. Please indicate how you implemented Lynch's method and what software was used. How did you obtain R2 values? Are these marginal R2 or conditional R2 values? Would results be different if you would have used a different approach (e.g. PGLS)?

Response: We have included more methodological details regarding the calculation of phylogenetic and species-specific variation in colour lightness using the R-package "ape" for Lynch's comparative method. The R-squared values represent the percentage of variation in raw colour lightness of species explained by the phylogenetic and species-specific components, respectively. Unlike the PGLS analysis, which controls for phylogenetic autocorrelation, Lynch's comparative method allows us to distinguish between the contribution explained by the phylogenetic relationship (phylogenetic signal) and the part that is specific to each species. It is essential to note that we do not consider the phylogenetic relationship as irrelevant; rather, we acknowledge that colour lightness evolution indeed contributes to the emergent pattern through diversification in relation to different mechanisms. Our findings underscore the significance of this phylogenetic signal, reflecting the regional prevalence of certain groups with adaptations to temperature and UVB in higher latitude realms, as well as pathogen resistance in lower latitude realms. Another advantage of using Lynch's comparative method for our analysis is the ability to discuss the effect of phylogeny in an assemblage-level context, which is not technically possible with other approaches, including PGLS.

Comment: 7. I am not convinced that partitioning analyses across realms should be given so much prominence in the main manuscript. While the results are important, I would suggest that the figures would be better placed in the suppl. material. This way you can free some space to depict patterns at the species level better.

Response: These results are important as they show for the first time that the importance of the three focal mechanism differs among biogeographical regions roughly along latitude. We now include more species-level results in the main text (Fig. 4), but would like to and need to - according to the journal guidelines- keep the realm-specific analysis in the text as they support our discussion. They do not qualify for the Supplementary Information because they are not just supporting additional technical validation that.

Comment: 8. In some places, I feel that the authors jump to conclusions too quickly, this is particularly the case when linking pathogen resistance to colour. If this interpretation has good support it would be good to explain it better, citing the appropriate references (e.g. is there anything known about links between severity to chytrid fungus and colour in amphibians?), if not make it clear that these are hypotheses that require further testing.

Response: To our knowledge there are no previous studies on the link between colour lightness and chytrid fungus severity currently available, which is not surprising given the low number of studies on the importance of colour in amphibians in general. However, also to strengthen the focus on species-level results we now provide an analysis of the severity and the interaction between colour and productivity (colour-based pathogen resistance) using data published in Scheele et al. 2019 and our data (Extended Data Fig. 2). We thank you for this suggestion and believe the results are addressing several important questions raised; They are support for the usefulness of productivity as a proxy for pathogen pressure, they are amphibian-specific support for the Gloger's rule and they are stressing the importance of incorporating trait-environment interactions into analysis of population declines.

Comment: Specific comments (Ln indicates line number)

Ln 23-25 This part may be selling the growing body of knowledge in this field a bit short. There have been many studies (some quite large scale) uncovering the links between coloration and climatic factors in ecto- and endotherms. A few examples: (Smith et al., 2016; Medina et al., 2018; Wisocki et al., 2019; Krah et al., 2019; Marcondes & Brumfield, 2019; Munro et al., 2019; Nicolaï et al., 2020; Delhey et al., 2021; Goldenberg et al., 2021; Kang et al., 2021; Idec et al., 2023).

Response: Thank you for the detailed examples. We now included several of these citations into our introduction.

Comment: Ln 58-60 there are other mechanisms associated with Gloger's rule, prominent among them is camouflage. Perhaps it would be good to mention these alternatives as well.

Response: Thank you very much for this comment. We now include this aspect more prominently into our introduction. We avoided an extensive discussion of camouflage as we cannot test for this biotic aspect of colour variation within the framework of our study. A sophisticated analysis of this aspect is provided in Loeffler-Henry et al. (2023). The approach to visually assess the species colour in comparison with the colour of the natural habitat and determining warning colours showcases that very different data would be needed for a discussion of signalling functions, as green frogs in the canopy and grass are camouflaged just alike brown frogs on brown soil. We only estimate the lightness of the species' skin.

Comment: Ln 87 I suspect the true r^2 value associated to the biological effects is lower, often these high r^2 values are mainly driven by the procedures used to account and control for spatial autocorrelation. Is there a possibility to separate them? How did you compute them? This also applies to lines 103-105.

Response: We fitted generalized additive models to account for spatial autocorrelation by including longitude and latitude as smoothed term (Table 1). The "raw" multiple linear regressions without accounting for spatial autocorrelation can be found in Table 1. The differences between the lms and gams (without/with accounting for spatial autocorrelation) are stated in the text. These differences result from the effects of latent environmental predictors (but see L131-134). We now clarify this in the text.

Comment: Ln 90 it would it be good to separate better assemblage-level analyses from species-level analyses. My preference would also be to first present the species-level analyses and then the assemblage-level analyses. As the latter is simply an emergent pattern from species-level data and distributions.

Response: For rhetorical reasons, it would be difficult to change the order of assemblage and species approach, as our main analysis is based on global patterns of colour lightness and the data collected has a focus on high spatial coverage rather than species coverage. However, to draw attention to the species approach we have included the coefficient plot of single linear regressions on the family level in the main part of the manuscript (Fig. 4) and also performed single linear regressions of all environmental predictors [Mean Annual Temperature (MAT), Elevation, Annual Precipitation (AP), Annual Enhanced Vegetation Index (EVI), the interaction of MAT and AP] and UV protection [Annual UVB radiation] at the species level (Supplementary Fig. 2).

Comment: Ln 109-118 From reading this section it is unclear whether these are analyses done at the species- or assemblage-levels.

Response: We now made clear that this section refers to the assemblage approach.

Comment: Ln 147 this is confusing, did you not refer to species-level data before (phylogenetic signal estimates require species-level data [how were these computed?])

Response: We moved the sentences in which we reported the phylogenetic signals in both species-level averages of environmental variables and colour lightness to the 'species-level analyses' section to improve clarity. This was included as motivation for applying Lynch's comparative method.

Comment: Ln 187-188 in birds the selection forces behind Gloger's rule are unclear, but camouflage may be playing an important role as well. I think that it would be good to expand on the potential mechanistic links between pathogen resistance and melanin pigmentation in anurans. In birds the mechanism is mainly concerned with plumage resistance to keratinolytic bacteria something that does not apply to anurans.

Response: In previous studies (e.g. Delhey et al. 2019), it was observed that species in areas with higher tree coverage tend to be darker, in addition to the influence of temperature and precipitation (a proxy for productivity). Initially, we did not consider tree cover as a relevant factor, assuming it was too simplistic to represent shade, a potential driver of camouflage. To test the effect of camouflage, detailed habitat information would be required (Loeffler-Henry et al. 2023), which may not directly translate into species in forested areas being darker. Moreover, vegetation like bushes could equally provide shade for ground-dwelling animals, such as most frogs and toads, regardless of whether trees are present. Birds might differ due to their mobility. Previous studies examining Gloger's rule utilized environmental proxies like precipitation, the interaction of temperature and precipitation, or productivity to assess pathogen pressure, as pathogens thrive under warm and wet conditions. We considered productivity as the primary proxy for pathogen pressure in testing Gloger's rule. In response to reviewer 2's suggestion, we performed an additional analysis (piecewise structural equation model, Supplementary Table 3) that incorporated tree cover. The results demonstrated that tree cover does not directly influence the colour lightness of anuran assemblages. Rather, productivity (EVI) more strongly drives tree cover, suggesting that causality may not be straightforward regarding the relationships of colour lightness with tree cover. Our findings indicate that pathogen pressure is more likely than camouflage to cause a Gloger's rule-like pattern in colour lightness. Previous observations of tree cover explaining part of the residuals in the relationship between colour lightness and the interaction of temperature and precipitation may be due to its additional role as a surrogate for productivity. Our measure appears robust to this effect, making it more suitable for testing Gloger's rule. We have also included literature support for the link between dark colouration and immunocompetence, evidence supporting Gloger's rule for non-bird species, and an additional analysis of the link between colour-based pathogen resistance and chytrid fungus severity.

Comment: Ln 199-200 revise writing.

Response: Done.

Comment: Ln 209-211 again, you are presenting pathogen-resistance as a fact, when in reality it is more of an hypothesis.

Response: We revised the sentence to underline uncertainty around the mechanism behind Gloger's rule.

Comment: Ln 219-221 I like this sentence!

Response: Thanks a lot!

Comment: Fig 2 could be included in the suppl. material.

Response: Our discussion relies on changes in the relative contributions of different drivers of colour variation both in terms of colour lightness biogeographical realm and the global scope of our data allows to understand these differences for the first time. All results essential to the reader need to be incorporated in the main text or Extended Data.

Comment: Ln 443-446 does this method yield similar conclusions to those obtained through the use of PGLS models? I think that it would be good to test this, since PGLS analyses are by far more common in the literature, and this would allow more targeted comparisons between taxa.

Response: PGLS analyses are commonly used in species-level studies to explore general effects among taxa by controlling for phylogenetic autocorrelation. However, this approach does not provide insights into which taxa follow the pattern and which do not, which is precisely our area of interest. For example, Dendrobatidae may deviate from ecophysiological colour patterns since their coloration primarily serves aposematism. Instead of merely accounting for phylogenetic signal, we seek to understand its outcome in trait variation. Lynch's comparative method offers a more suitable approach as it decomposes a trait into its phylogenetically predicted and species-specific variation. This not only sheds light on the interplay of trait evolution and emergent patterns but also disentangles the impact of such signals from non-evolutionary causes in assemblage-level analysis. Rather than treating phylogenetic autocorrelation as a bias (as in PGLS), our method informs about the influence of distributional and diversification success of certain lineages (P-component or family differences at the species-level and P-component at the assemblage-level). We have now clarified this in the 'results' section.

Comment: Ln 449-450 how did you assess whether this approach effectively accounted for spatial autocorrelation? Plots, Moran's I, etc.?

Response: Thank you. We now include correlograms for the residuals of ordinary least-squares models and those of generalized additive models that included and additional spline-based smoothed term of geographical coordinates (Supplementary Figure 4). This comparison shows that our approach reduced the spatial autocorrelation in the model residuals to a minimum.

Comment: Ln 477-479 I feel that you are too assertive here, the data you generated does not allow you to say this, it simply indicates whether variation in colour lightness is higher or lower than expected. Whether this is caused by a dominant driver or multiple drivers or something else is not

something that you can determine with these data. While you can speculate that this is the case in the discussion here it sounds to clear-cut.

Response: Analysis of trait over- or under-dispersion are commonly used to understand the extent to which environmental factors filter for functionally more similar species. In the case of over-dispersion there are two different possible interpretations. At a fine spatial grain where species are more likely to interact with one another and for traits that are associated with biotic interactions such as resource competition, niche overlap or trophic position overdispersion is commonly interpreted as competition at a coarse grain, as in our case, overdispersion should indicate the presence of multiple competing factors. We now acknowledge the uncertainty with the interpretation of over-dispersion in this paragraph.

Comment: Extended Data Fig. 1 what is the reference for the colour wheel used?

Response: The colour wheel was developed by Stefan Pinkert and inspired by Bishop et al. 2016 where they also used a predefined colour palette to determine the colouration of ant assemblages (now Extended Data Fig.3).

Comment: Extended Data Fig. 4 This figure is very hard to follow, I think these data should be presented as a table.

Response: We enlarged the figure but would like to keep it as visual of the effects because a table would decrease the overview, i.e. show which of the four predictors was significant/ indicate its direction/ and hypothesis support. This is how we want to draw more attention to the species-level results, as you suggested in your previous comments. For now, we would like to present the results as a figure (Former Extended Data Fig. 4, now Fig. 4) but below we provide it as a table as suggested by you. Given that the table is very large we are uncertain if this presentation is preferable. We would hence ask for your advice on what presentation should be used.

Table 2. Environmental drivers of the colour lightness of anuran species (n = 2,984).

Coefficients, standard errors, and standardized effect size of single linear regression models of colour lightness and environmental variables.

Family	MAT		Elevation		EVI		UVB	
	slope±SE	t	slope±SE	t	slope±SE	t	slope±SE	t
Alsodidae	10.35±20.16	0.51	-2.90±5.66	-0.51	-2.54±7.33	-0.35	3.51±12.88	0.27
Alytidae	34.00±46.84	0.73	-4.84±27.55	-0.18	-6.22±24.20	-0.26	-11.52±17.18	-0.67
Aromobatidae	-34.98±9.92	-3.53	-9.28±8.65	-1.07	-22.64±4.77	-4.74	-32.72±8.85	-3.70
Arthroleptidae	5.07±4.14	1.22	2.19±3.22	0.68	-5.23±2.63	-1.99	2.71±3.38	0.80
Batrachylidae	-9.53±23.62	-0.40	-10.68±11.34	-0.94	25.43±17.11	1.49	-9.02±13.25	-0.68
Brachycephalidae	0.74±10.19	0.07	7.58±14.22	0.53	-5.80±4.02	-1.44	-1.79±25.15	-0.07
Brevicipitidae	-4.96±9.78	-0.51	-3.73±5.93	-0.63	-14.06±6.92	-2.03	-2.89±7.05	-0.41
Bufoidea	-0.83±3.17	-0.26	0.09±1.47	0.06	-11.39±1.72	-6.64	-1.83±2.39	-0.77
Centrolenidae	55.46±6.53	8.49	20.73±3.21	6.46	19.21±2.91	6.60	41.45±4.88	8.49
Ceratophryidae	-26.45±13.53	-1.96	13.12±11.79	1.11	-26.62±9.21	-2.89	-16.06±15.02	-1.07
Craugastoridae	-11.73±3.74	-3.14	-2.11±1.16	-1.82	-12.22±1.68	-7.28	-6.06±2.14	-2.83
Cycloramphidae	-36.16±11.46	-3.16	-28.17±18.58	-1.52	-21.94±4.58	-4.79	-49.13±27.74	-1.77
Dendrobatidae	-29.56±4.73	-6.25	-7.71±2.20	-3.50	-20.78±2.10	-9.88	-20.13±3.34	-6.02
Dicroglossidae	-12.97±5.72	-2.27	0.22±2.22	0.10	-15.19±3.77	-4.03	-18.36±6.07	-3.03
Eleutherodactylidae	-3.69±10.58	-0.35	-1.83±11.34	-0.16	-13.01±6.37	-2.04	-8.74±12.46	-0.70
Hemiphractidae	20.63±9.89	2.09	1.23±3.71	0.33	1.60±4.28	0.37	5.36±6.43	0.83
Hylidae	14.71±2.87	5.12	-1.52±2.82	-0.54	0.62±1.40	0.45	9.35±2.89	3.24
Hylodidae	-38.71±10.01	-3.87	-30.26±11.74	-2.58	-21.57±3.89	-5.55	-51.32±20.21	-2.54
Hyperoliidae	38.82±3.62	10.71	14.24±2.75	5.18	16.53±2.47	6.70	30.50±3.03	10.06
Leptodactylidae	-7.52±4.11	-1.83	-7.09±4.59	-1.54	-12.22±2.13	-5.73	-7.30±4.44	-1.65
Limnodynastidae	16.95±8.60	1.97	-1.19±27.69	-0.04	-12.42±8.18	-1.52	17.74±11.83	1.50
Mantellidae	-11.40±4.36	-2.62	-11.60±4.21	-2.76	-14.37±2.31	-6.23	-16.68±5.02	-3.32
Megophryidae	-11.33±5.44	-2.08	-8.01±1.45	-5.52	-20.02±2.60	-7.70	-17.79±4.69	-3.79
Microhylidae	-12.02±3.50	-3.44	-9.85±4.89	-2.02	-14.94±1.98	-7.55	-16.42±3.80	-4.32
Myobatrachidae	9.64±6.13	1.57	-7.64±15.61	-0.49	-9.79±5.38	-1.82	9.50±7.06	1.35
Odontophrynidae	-15.96±11.26	-1.42	-33.14±21.33	-1.55	-15.99±5.62	-2.85	-15.43±15.92	-0.97
Pelodyadidae	27.50±5.77	4.77	-16.63±16.63	-1.00	16.54±5.79	2.86	18.07±6.60	2.74
Petropedetidae	-3.76±12.85	-0.29	7.42±8.72	0.85	-8.67±8.77	-0.99	2.24±9.64	0.23
Phrynobatrachidae	13.69±5.42	2.53	3.03±3.91	0.77	-1.71±3.74	-0.46	9.96±4.48	2.22
Phyllomedusidae	7.25±6.69	1.08	13.56±9.69	1.40	-2.52±3.27	-0.77	9.17±7.91	1.16
Pipidae	-12.25±6.98	-1.76	-7.69±5.15	-1.49	-14.29±4.09	-3.49	-12.43±5.85	-2.13
Ptychadenidae	19.73±6.30	3.13	5.62±3.45	1.63	5.31±4.96	1.07	12.26±4.55	2.70
Pyxicephalidae	23.6±67.61	3.11	7.64±3.41	2.24	-1.76±5.91	-0.30	16.01±5.66	2.83
Ranidae	-9.25±4.04	-2.29	1.98±1.76	1.13	-12.12±2.35	-5.15	-7.85±3.48	-2.26
Rhacophoridae	5.93±4.02	1.48	12.97±3.01	4.31	-1.86±2.13	-0.87	6.30±4.11	1.53

Note: MAT = Mean annual temperature, EVI = Mean annual EVI, UVB = Mean annual UVB. Predictors with significant effects at $p < 0.001$ are highlighted.

References

- Delhey, K., Dale, J., Valcu, M. & Kempenaers, B. 2021. Migratory birds are lighter coloured. *Curr. Biol.* 31: R1511–R1512.
- Goldenberg, J., D'Alba, L., Bisschop, K., Vanthournout, B. & Shawkey, M. 2021. Substrate thermal properties influence ventral brightness evolution in ectotherms. *Commun. Biol.* 1–10. Springer US.
- Idec, J.H., Bishop, T.R. & Fisher, B.L. 2023. Using computer vision to understand the global biogeography of ant color. *Ecography (Cop.)*. 2023(3): e06279.
- Kang, C., Im, S., Young, W., Yunji, L., Fox, D.S.- & Huertas, B. 2021. Climate predicts both visible and near- infrared reflectance in butterflies. 1869–1879.
- Krah, F., Büntgen, U., Schaefer, H., Müller, J., Andrew, C., Boddy, L., et al. 2019. European mushroom assemblages are darker in cold climates. *Nat. Commun.* 10: 2890. Springer US.
- Marcondes, R.S. & Brumfield, R.T. 2019. Fifty shades of brown: Macroevolution of plumage brightness in the Furnariida, a large clade of drab Neotropical passerines. *Evolution (N. Y.)*. 73: 704–719.
- Medina, I., Newton, E., Kearney, M.R., Mulder, R.A., Porter, W.P. & Stuart-Fox, D. 2018. Reflection of near-infrared light confers thermal protection in birds. *Nat. Commun.* 9: 3610. Springer US.
- Munro, J.T., Medina, I., Walker, K., Moussalli, A., Kearney, M.R., Dyer, A.G., et al. 2019. Climate is a strong predictor of near- infrared reflectance but a poor predictor of colour in butterflies.
- Nicolai, M.P., Shawkey, M.D., Porchetta, S., Claus, R. & D'Alba, L. 2020. Exposure to UV radiance predicts repeated evolution of concealed black skin in birds. *Nat. Commun.* 11: 2414.
- Schwalm, P.A., Starrett, P.H. & McDiarmid, R.W. 1977. Infrared reflectance in leaf-sitting neotropical frogs. *Science (80-)*. 196: 1225–1226.
- Smith, K.R., Cadena, V., Endler, J.A., Porter, W.P., Kearney, M.R. & Stuart-Fox, D. 2016. Colour change on different body regions provides thermal and signalling advantages in bearded dragon lizards. *Proc. R. Soc. B Biol. Sci.* 283: 1–9.
- Wisocki, P.A., Kennelly, P., Rivera, I.R., Cassey, P., Burkey, M.L. & Hanley, D. 2019. The global distribution of avian eggshell colours suggest a thermoregulatory benefit of darker pigmentation. *Nat. Ecol. Evol.* doi:10.1038/s41559-019-1003-2.

Reviewer #2 (Remarks to the Author):

General comment: I have now read with interest the manuscript titled ‘Frogs reveal the global importance and interplay of colour-based protective and thermoregulatory functions’. Laumeier et al. set out to understand the drivers of assemblage-level colour lightness in frogs using data from 40% of all species. Specifically, they investigated three main drivers of colour lightness – thermoregulation, pathogen resistance, and UVB protection – using appropriate environmental proxies. Laumeier et al. show color lightness to decrease in colder regions and regions with higher pathogen pressure and UVB radiation.

I enjoyed reading the paper, and I think it’s a topic that will be of broad interest. However, I cannot recommend the manuscript for publication in its current form. I will explain this in detail below.

General response: Thank you for this positive and constructive feedback. We have carefully considered all your comments and made revisions accordingly. The supplementary analyses and references you suggested underlined the robustness of the results and strengthened our conclusions. We are very grateful for the time you have invested to improve our manuscript.

Note that the line references in comments are from the original document, whereas the line references in our responses refers to the revised version of the manuscript.

#####

Comment: First, the study quantified colour lightness using a standardized colour wheel. This was achieved by visually matching photographs from field guides by a single author (L.B.). While visual estimation by human observers has its advantages (easily quantifiable for a large number of species), this method does not capture variation in the other spectrum – ultraviolet or infrared (which together account for >50% of solar radiation). The dataset is thus based on human perception of colours (visible range).

Recent studies have shown infrared reflectance may also play a critical role in heat gain (e.g., Medina et al. 2018 and also Schwalm et al. 1977). I would like the authors to address this issue by running separate analyses by using data from actual reflectance values when available from the literature. For example, see Clusella-Trullas et al., 2008 who collected reflectance data from literature sources to test the thermal-melanism hypothesis in lizards. I think a similar analysis can be conducted for a few species to support the main claims in this study or at least show color lightness measured from different methods is tightly related. In any case, the authors must better explain and provide justification for using human-based classifications (only visible spectrum) and make readers aware of the limitation.

Schwalm, P.A., Starrett, P.H. and McDiarmid, R.W., 1977. Infrared reflectance in leaf-sitting neotropical frogs. *Science*, 196(4295), pp.1225-1226.

Medina, I., Newton, E., Kearney, M.R., Mulder, R.A., Porter, W.P. and Stuart-Fox, D., 2018. Reflection of near-infrared light confers thermal protection in birds. *Nature communications*, 9(1), p.3610.

Clusella-Trullas, S., Terblanche, J.S., Blackburn, T.M. and Chown, S.L., 2008. Testing the thermal melanism hypothesis: a macrophysiological approach. *Functional ecology*, pp.232-238.

Response: Thank you for the valuable input. We have now included several studies to support the notion that the visible spectrum of colours we assessed is representative of the overall ability of heat gain. We also revised the statement about the limitations of our approach accordingly and provided a comparison of measurements based on different methods (Extended Data Fig. 4). It is important to note that literature data on the reflectance of anuran skin colours is limited and often does not focus on the thermoregulatory body part (dorsal) or account for colour coverage. In support of our approach, previous studies on butterflies (assessed in the visual spectrum based on illustrations or photos) and yeast have demonstrated a strong correlation between colour estimates and species' body temperature ($r = -0.76$; Zeuss et al. 2014) and the temperature access rate of yeast (Cordero et al.

2018). Recently, a study in frogs has also provided further support for this relationship (Park et al. 2023). Although we cannot experimentally test this relationship in frogs within the scope of our study, evidence from other ectotherms, including some frog species, aligns with the findings of Clusella-Trullas et al. (2008) and suggests that colour lightness assessed in the visible spectrum is a suitable surrogate for overall reflectance.

Comment: Second, I thank the authors for showing interobserver differences in color lightness classification (Extended data Fig.2). However, there is only a moderate correlation between the two observers ($R^2 = 0.65$), suggesting there is still subjective bias in rating color lightness. To make the argument much stronger, the authors can directly measure color lightness from the scanned photographs (pixel level) for a few species and compare them with lightness classified by human observers. This can be done by extracting pixel colour value manually from images using software such as ImageJ (see Roberts et al. 2022 for a similar approach). I suggest authors show this objective measure of color lightness results in the supplement.

Roberts, S.M., Stuart-Fox, D. and Medina, I., 2022. The evolution of conspicuousness in frogs: When to signal toxicity?. *Journal of Evolutionary Biology*, 35(11), pp.1455-1464.

Response: Please note that the goodness of fit is given as r-squared value and not - as maybe more commonly done - as Pearson correlation coefficient ($r = 0.81$; i.e. highly correlated). We now use this measure and adapted the Figure. The remaining variation can have different sources such as an inclusion of shaded/shiny parts of the skin, a slightly different perception of the colour or a different estimate of the coverage of a specific colour. This should improve with more training of the observers – person B was ‘naïve’. Note that in any case errors would have inflated the false-negative probability and hence not our interpretation of significant results.

We used the suggested methods from Roberts et al. (2022) to extract the colour of a small set of pixels of each distinct colour (same as for the human-based estimation) from scans for 107 species. We used the same coverage weight for both set of colours. This comparison showed that the colour lightness estimates based on the colour extraction and human-based assessment are highly similar ($r = 0.80$). We added this comparison to the Extended Data (Extended Data Fig. 4). Note that particularly for lower colour lightness values the comparison shows that the human-based estimates are higher than those of the colour extraction, which likely stems from the overall darker colour of the scans compared to the print. Also, part of the difference between the two approaches may result from the difficulty to probe (extract) a pixel that has a representative colour for a larger part of the body.

Comment: Third, the statistical analysis must be improved. It is unclear how authors accounted for multicollinearity in their analysis. I could not find details in the methods section. It is well known that temperature and UV radiation are tightly correlated with elevation, and productivity is related to temperature. Consequently, the effect sizes and direction of regression co-efficient presented in the ms are not reliable. Therefore, any inference made on the contribution of each variable in explaining lightness value (for e.g., Lines 28-30) is potentially incorrect.

Further, these environmental predictors may or may not act in synergy (see Delhey 2019 review) – which was not addressed in the current manuscript. To test if these predictors are acting independently or in synergy the author could use a Structural Equation Modeling analysis and see if the path of influence between predictors and response variable is independent or not. There are R packages available for SEM methods (‘lavaan’ or ‘phylopath’). I strongly suggest authors present their results in a path-analysis framework.

Response: In the overall regression model, variance inflation factors indicate high multicollinearity between temperature and UV radiation (Bio1: 10.3, Elevation: 2.6, EVI: 1.2, UV-radiation: 10.0). Removing UV radiation significantly reduces variance inflation scores (Bio1: 1.2, Elevation: 1.1, EVI: 1.2). Accounting for spatial autocorrelation further reduced multicollinearity among environmental predictors.

To isolate the part that is not multicollinear among different drivers, we employed hierarchical partitioning analysis, providing independent contributions of all drivers (Fig. 2 and 3). While we acknowledge the strong synergistic effect of temperature and precipitation on colour lightness (related to thermoregulation and pathogen resistance, see Delhey et al. 2019), our supplementary path analysis show that this is not the case for temperature and productivity as measured by EVI (Table 3).

This path analysis showed that warmer regions are more productive, but productivity has a negative effect on colour lightness, contrasting the direct positive effect of temperature. Hence, the indirect effect of temperature on colour lightness via productivity opposes its direct effect. The independent effects are much weaker than the direct effects (AMT: direct = +0.978 vs. indirect = -0.088) and show similar patterns for UVB radiation (UVB: direct = -0.426 vs. indirect = +0.025).

Comment: Fourth, another major issue with the ms is attributing the effect EVI on lightness to pathogen pressure alone. Productive environments are likely associated with closed habitats (i.e., habitats with low light levels, such as high canopy trees especially in the tropics), where animals are predicted to be darker in coloration, potentially due to crypsis or other reasons. Therefore, the fact that authors find a strong effect of EVI on lightness could be due to the selection for signaling based on light conditions (in addition to pathogen?). See the below study by Marcondes et al. 2021 who showed forested birds have darker plumage even after controlling for the climate. See also Delhey et al. 2019 who used the percentage of tree cover as a predictor.

I strongly suggest authors include a proxy for the light environment (tree cover or forested habitat type) as a predictor (see below papers). As suggested above, incorporating this in a path analysis framework will enable authors to tease apart the main driver of lightness in the tropics.

Marcondes, R.S., Nations, J.A., Seeholzer, G.F. and Brumfield, R.T., 2021. Rethinking Gloger's rule: climate, light environments, and color in a large family of tropical birds (Furnariidae). *The American Naturalist*, 197(5), pp.592-606.

Delhey, K., Dale, J., Valcu, M. and Kempenaers, B., 2019. Reconciling ecogeographical rules: rainfall and temperature predict global colour variation in the largest bird radiation. *Ecology Letters*, 22(4), pp.726-736.

Response: We used tree cover data from EarthEnv in a structural equation framework to test your assumption (Supplementary Table 3). The results of this analysis show that tree cover has no direct effect on colour lightness. In addition, by including an unresolved causal relationship between productivity and tree cover we show that both are strongly correlated but that tree cover is driven by productivity rather than the other way around (+0.703 vs. +0.615). We conclude that tree cover has no additional effects on colour lightness and that crypsis related to shading by trees does not affect our results.

Comment: Finally, I found the results section disconnected from the introduction. For instance, the authors directly present the results from regression analysis where environmental predictors are used as a proxy to test the three main drivers of lightness. A reader may find it hard to connect these results with the main mechanisms in the introduction without reading the methods. I suggest authors include a brief aim and the method employed to address them at the start of each results section (1 or 2 sentences).

I also find that many key methodological details are not described in the methods section (see specific comments below). For example, no justification was provided on why specific environmental variables were chosen and the exact mechanism linking that variable with the three main drivers. I understand the journal restrictions on the word limit. However, I suggest authors include detailed methods with all the necessary details as done in a normal full-length paper in the supplement.

Response: At several parts of the methods, we added more details, and the specific comments were very helpful to understand where more detail was needed. For the paragraph 'Environmental data' in the 'methods' section however, we are uncertain how to adapt: 'Based on predictions of the thermal

melanism hypothesis (colour-based thermoregulation), Gloger's rule and the UVB protection hypothesis, four environmental proxies for temperature (mean annual temperature (v2³⁵), elevation³⁶), productivity (mean annual EVI³⁷), and UVB radiation³⁸ were derived from global high-resolution data.' However, we now include this reasoning also at the beginning of the 'results' section to clarify the why we have chosen each environmental variable, as early as such detail can appear with the journal format.

#####

Specific comments

1.Line 56: Also known as 'Bogert's rule'.

Response: Done.

Comment: 2.Line 58: The below study might be highly relevant.

Fu, T.T., Sun, Y.B., Gao, W., Long, C.B., Yang, C.H., Yang, X.W., Zhang, Y., Lan, X.Q., Huang, S., Jin, J.Q. and Murphy, R.W., 2022. The highest-elevation frog provides insights into mechanisms and evolution of defenses against high UV radiation. *Proceedings of the National Academy of Sciences*, 119(46), p.e2212406119.

Response: We now include this study.

Comment: 3.Line 59: 'darker species tend to be more resistant to pathogen' – the mechanism should be added here. Gloger's rule was proposed for endotherms, and it has not been shown in ectotherms (at least in amphibians and reptiles). Consider adding citations for ectotherms backed up by clear mechanistic explanations.

Response: Melanin enhances the structural integrity of cells and is associated with an enhanced immunocompetence, whereby a higher melanin concentration in the cuticular (darker skin coloration) should prevent the penetration by fungal and bacterial pathogens. We rephrased this sentence to highlight the underlying mechanism and added support for reptiles (Delhey et al. 2019 and sources therein; see also Becker et al. 2012). Whereas the resemblance of reptilian and amphibians skin characteristics suggest that colour-based pathogen protection also applies to the latter we could find only indirect experimental support for this taxon (melanin enhances immunocompetence - the frequency of immune cells; see Franco-Belussi et al. 2016). We now also provide an additional analysis using data on the severity chytridiomycosis for anuran species. In line with the Gloger's rule, we show that lighter species in productive regions have a greater severity to chytridiomycosis. This was not the case for colour or productivity alone. We added this analysis to the main text (Extended Data Fig. 2) and contextualize its results.

Franco-Belussi, L., Sköld, H.N., de Oliveira, C. Internal pigment cells respond to external UV radiation in frogs. *J. Exp. Biol.* **219**, 1378–1383 (2016).

Comment: 4.Line 60: Not only parasites but also for signalling (e.g., crypsis). See my general point #4.

Response: At this point in the introduction, we refer to physiological functions of colour in ectotherms. We believe that an earlier discussion of the importance of crypsis and other colour-based functions could be confusing because we focus on colour lightness not on colour in general and test for colour-based physiological trait-environment rather than biotic interactions. Testing for crypsis and aposematism would require a different approach (e.g. species colour habitat colour contrasts; see e.g. Loeffler-Henry et al. 2023).

Comment: 5.Line 68: The authors did not directly test any of the three drivers. Consider stating that you tested these mechanisms with environmental proxies.

Response: We now clarify that with our macroecological (phenomenological) approach we cannot test a mechanism itself, but only for the effect and generality of drivers that are associated with experimentally well-supported mechanisms. We use the following stringent nomenclature driver -> mechanism -> pattern. For instance, temperature -> colour-based thermoregulation -> darker were colder. Should we rephrase our text to use 'environmental proxies of mechanisms' instead of 'drivers' and refer to 'drivers' instead of 'mechanisms'?

Comment: 6.Line 84: 'increasing temperature...and decreasing productivity' this could simply be due to multicollinearity. See general comment #3. This also applies to realm-level analysis that highlights the importance of each predictor.

Response: The above-mentioned variance inflation analysis demonstrates that temperature and productivity are least collinear ($vif < 2$). We find that at the realm-level the effects of both drivers on colour lightness can be both convergent and divergent. Temperature and productivity are positively correlated. The hierarchical partitioning results in Fig. 2 and 3 are independent contributions and highlight that there are strong independent effects of temperature and productivity.

Comment: 7.Line 110: 'generally weaker but remained robust'. It is unclear whether if authors standardized their regression coefficients to make such a comparison across the methods. Clarify and add details in the methods section.

Response: The results are directly comparable. The predictors are scaled and centered (z-scaled). The only difference is an additional trend-surface term (spline based smoothed term of geographical coordinates). We rephrased the sentence to improve clarity.

Comment: 8.Line 176: But according to recent studies on birds (general comment #4), the influence of signalling strategies on colour evolution cannot be ignored. See the below review. Rojas, B., 2017. Behavioural, ecological, and evolutionary aspects of diversity in frog colour patterns. *Biological Reviews*, 92(2), pp.1059-1080.

Response: We added this very useful review and specify that this statement was referring to anurans. We do not think that signalling is irrelevant but cannot directly address its effect within the framework of our study. Note that our focal measure is colour lightness and not colour and that this proxy of melanisation does not allow to discuss colour patterns. We are aware of the fact that multiple other functions, particularly aposematism and crypsis may have increased the scatter in the documented relationships but an adequate discussion of this sources of bias would need data on the habitat context in reference to the specific colour of the frog species (e.g. brown frog on brown soil), which is outside the scope of your study. Note that these additional functions could have increased the probability of detecting a false-negative result and but cannot have affected our interpretations of significant effects.

Comment: 9.Line 185: I am not entirely convinced by the structural integrity idea in Anurans. The cited examples are for birds and ants, which probably make sense due to external structures (feathers and exoskeletons).

Response: We added support for this claim from a review of several relevant studies on reptiles and amphibians (Delhey et al 2019). Delhey also notes "*One such plausible selection force could be an increase in the prevalence of parasites and pathogens in more humid or vegetated environments ([...] Becker et al., 2012).*". We now also cite Becker et al. (2012) as support for this claim in anurans.

Comment: 10.Line 376: The 2021 version of IUCN has data for about ~6700 species, I always assumed colour lightness values were hard to obtain relative to distribution. I suggest authors provide family-level data coverage for both in the supplement.

Response: Done. The included literature sources included several species that are newly described but there were also many species for which no range maps were available (e.g. *Pristimantis* in Equator). We contacted the authors to ask for occurrence records to potentially include more species but without success. We now provide the family data coverage for both in Supplementary Table 4.

Comment: 11.Line 385: Are the images photographs of actual animals or illustrations? If from illustrations, the colours may not be accurate (drawn by artists). Is it possible to show that there is no difference between the two sources?

Response: The vast majority of the colour lightness estimates are based on *in situ* pictures, because specimens fade in colour after their death and because drawings are only limitedly available. We used drawings for only for Australian and European species pictures because they were of higher quality and more standardized regarding the light conditions. Note that most of the literature on large-scale patterns in colour lightness is based on drawings from field guides (e.g. Zeuss et al. 2014, Pinkert et al 2017, Heidrich et al. 2018, Delhey et al. 2019). In previous studies, we found similar spatial patterns for North American (based on images) and European butterflies (based on drawings, Stelbrink et al. 2018). We now added a comparison of the colour lightness estimates of the subset of species from Australia (Extended Data Fig. 4) but note that the pictures have had a poor quality and were red-biased, needed to be scanned and were partly different colour variants. The Pearson correlation coefficient for this comparison was 0.65 ($P < 0.001$, $n = 112$).

Comment: 12.Line 389: 5% as in saturation levels?

Response: The saturation of a colour ray in the wheel is provided in for variants (Extended Data Fig. 1). The four steps are the same for each colour comprising 10, 40, 70 or 100 % in saturation. The coverage, however, was assessed in 5% steps (any value of a sequence of 5, 10, ..., 100).

Comment: 13.Line 389-390: Consider proving the formula for the weighted mean. Also mentioned this index as colour-lightness to avoid confusion (same as what authors refer to as melanisation?)

Response: We now included the formula in the methods section: Colour lightness = $\frac{((R_1 + G_1 + B_1)/3) \cdot \text{coverage}_1 + ((R_2 + G_2 + B_2)/3) \cdot \text{coverage}_2}{\text{coverage}_1 + \text{coverage}_2}$. For an example calculation see Extended Data Fig. 3). Red, Green, and Blue are the colour channels of the respective colour values (Supplementary Table 2) of the colour wheel (Extended Data Fig. 1). Indices the above formula (here for two colours: 1 and 2) indicate a specific colour and its coverage assessed for an individual.

Comment: 14.Line 393: Consider proving the total number of images analysed and the average number of images per species.

Response: Our analyses included 4,305 individuals of 3,059 species (1.4 individuals on average). We now added the latter values to the text. Note that we focussed our assessment on maximizing the number of species rather than morphs/subspecies but included estimates whenever they were provided.

Comment: 15.Line 412: I am not sure on what basis the authors used a 50% cut-off – consider adding relevant references or running sensitivity analysis for other cut-off values.

Response: This absence/presence cut-off is a common compromise of the resolution of expert range maps and the study specific scale. The resolution of expert range maps (polygons) are technically

infinitesimally fine. However, there is considerable uncertainty around their extent which is addressed by using a rather coarse resolution for the grid at which the distributions are resampled to. In any case the intersection of a grid and species ranges will result in fragments of different size and the only way to address this is to assume a cut-off. A cut-off of 100% will drop those grids as presences that is occupied almost entirely and still would not address the issue of spatial uncertainty. We are unaware of a study that evaluated this cut-off and could provide a guideline how to measure accuracy in the case of biogeographical patterns. Due to the nature of intersections of a stationary grid and geographical varying species ranges a cut-off should be species-specific rather than taxon- or dataset-specific. However note that the species-level environmental estimates are aggregated values for environmental predictors (1 km resolution) across the range of species.

Comment: 16.Line 412: Regarding the above point, it is unclear how the authors treated narrow-ranged species (1-2 grid cells). Most of these species' ranges do not cover the entire grid cell (usually, endemics have <50% coverage). Did the authors remove all such species from the analysis? Please clarify.

Response: Thank you for pointing out this mistake. This criterion has not been used for the final analysis and we now remove it from the methods. Of the species for which colour data was available only two species were removed from the combined dataset of colour and distribution data due to the water coverage criterion. We did not exclude narrow species that only covered one to two grid cells.

Comment: 17.Line 448: I could not confidently conclude that there is no spatial autocorrelation without a test on residuals (Moran's I?). Please provide test results in the supplement.

Response: We now provide correlograms for the residuals of initial models and those controlled for spatial autocorrelation in the supplement (Supplementary Fig. 4).

Comment: 18.Line 493: Should this be the proportion of individuals per family rather than absolute numbers?

Response. We excluded families with fewer than 10 (not 10%) species.

Comment: 19.Line 499 to 500: It is unclear from where the authors obtained the activity data from. Consider citing the source reference.

Response: Thank you for highlighting this mistake. We now added the respective sources to the text, which were the same field guides as we used for the colour lightness assessment. The data for Africa was supplemented with pictures for additional 175 species by the author of the respective field guide (Mark-Oliver Rödel, co-author on this paper).

Comment: 20.Line 1: Minor point. The title of the paper can be changed to 'The global importance and interplay of colour-based protective and thermoregulatory functions in frogs'. I also suggest replacing the mechanisms with actual predictors (e.g., 'thermoregulatory' with 'temperature') in the title, as done in Delhey et al. 2019.

General response: We have changed the title to your first suggestion. Thank you for this suggestion. By using the predictor names, the title would become too long and stating the mechanisms was our way to highlight the direction of effects in addition to their significance ('shape', 'predict', 'determine').

REVIEWER COMMENTS

Reviewer #1 (Remarks to the Author):

I thank the authors for their revision and for taking into consideration my previous suggestions. While the manuscript has improved, I do still have some concerns/suggestions as detailed below.

Species-level analysis vs assemblage-level analyses:

In my previous review I suggested to make this level of analysis the main level of analysis simply because this is closer to the level at which selection is happening. The authors counter this suggestion by stating in their response letter that:

(1) assemblage-level analyses are more relevant to their questions because "biogeographical patterns, differences in the relative importance of drivers among the biogeographical realms and the colour lightness diversity are purely spatial". Given the high levels of endemism in anurans, it is eminently possible to carry out realm-specific analyses at the species level (just include all species found in each realm, even if there may be a few cases where the same species pops up in different realms). While it is true that lightness diversity can only be studied at the level of the assemblage, these results are not particularly prominent nor relevant to the main message of the manuscript. As a matter of fact, I could not find any reference to lightness diversity in the Discussion. I suggest that they could be easily shifted to the suppl. material or dropped from the manuscript altogether.

(2) "data was quantified from literature sources with a geographical scope, so that the strength of our data is the high spatial coverage rather than the species coverage". I do not quite understand why this is the case, all estimates of colour were averaged obtaining a single value per species, thus there is no spatial component to coloration, other than the distribution of each species. In sum, I am not convinced that shifting the focus from assemblage to species-level analyses would be so disruptive or change the scope of the paper. After all, when you discuss the implications of your research these refer to the capacity of species to cope with changes.

In addition, currently the species-level analysis focuses on presenting results at the family-level, but the reason for this is unclear to me: For example, in Ln 175 the authors argue that given the phylogenetic effects this "therefore" implies that estimates need to be computed separately for each family. I cannot follow the logic here. While such analyses may allow us to see whether there are some differences between taxonomic groups, I would like to see an overall species-level analysis that would include all species for which there is data (now you exclude families with fewer than 10 species). Please include a simple model with main effects only, so that these results can be compared with that of other studies. Family-specific effects can be moved to the online material.

It still remains unclear to me how phylogenetic relatedness was included in the species-level analyses. Was this done using Lynch's method? Please clarify and provide full results in online suppl. table. I suggest that a simple pgl's with lambda correction may be the most suitable alternative here.

Links between pathogen-resistance and colour:

The authors now include a very interesting comparative analysis testing for an association between lightness and resistance to fungal infection (arguably the main pathogenic threat to anurans). However, more information on how these analyses were carried out is needed, as the information in Ln 618-623 is too limited.

- (a) How did you define co-occurring species?
- (b) What was the response variable in the analysis (to find this one has to dive into the extended data)?
- (c) Why only include productivity of all climatic predictors?
- (d) How well documented is the link between productivity and pathogen pressure assumed here?

(e) Again, families with <10 species are excluded, why? Given the limited sample size you should strive to include as many species with data as possible. I do not see the logic to run the main analysis separately per family.

(f) Why is the interaction between productivity and lightness the relevant test here? Please add analyses without interaction term as tables to the suppl. material as well.

(g) What kind of model was used for this analysis, Lynch's method? Please clarify. As above, I suggest that a simple ppls with lambda correction may be the most suitable alternative here.

Presentation of results:

I find that the main statistical results as presented in Fig2b, Fig 3b and Fig 4 are very hard to follow. Effects for all predictors are shown on the same graph making it very difficult to recognize overall patterns. One possibility for improvement would be to present forest plots for each predictor in a separate panel.

In lines 140-142 you state "Hence, accounting for idiosyncrasies of the responses among realms with different biogeographical histories largely removed the effect of latent spatially autocorrelated variables" Why is this the case? What evidence was used to support this statement?

Reviewer #2 (Remarks to the Author):

General comments

I have now gone through the revised manuscript. The authors have done a wonderful job of responding constructively to the suggestions both reviewers provided. In the revised version, the authors have broadened the introduction, included an analysis of possible links between color lightness and pathogen resistance, as well as a path analysis testing the role of tree cover on lightness. The authors also performed a separate analysis comparing color lightness measured by two methods (color wheel vs. pixel approach). This new set of analyses strengthens the author's approach and their previous conclusions that anuran color lightness is primarily associated with four main drivers – temperature, elevation, productivity, and UVB radiation. I have a list of minor suggestions below (some are stylistic) that authors may want to consider. Nice work!

Specific comments

1. Line 87 to 96: This entire paragraph in the introduction has a bunch of results and inferences before the actual results! I could not locate the reviewer's comments requesting such an addition to the introduction. I strongly feel that it doesn't belong here. Consider moving it to the discussion (see another alternative below).

2. Line 99 onwards: Consider moving this new first paragraph of results to the introduction (as the last paragraph). This nicely provides background and predictions about the study.

If the authors decide to keep the texts in Lines 87-96, then these texts (99-104) can be moved before the part that provides brief results and significance of the study (currently lines 87-96).

3. Line 188: Thanks for this new addition – very interesting results. The authors present standardized effect sizes of the interaction between color lightness and productivity on chytridiomycosis severity (ED Fig.2). It is important to note that only 9 out of 20 families show a positive effect. It is unclear about the effect sizes and significance of the interaction term in the global model. I suggest authors provide the full model results as a table in the supplement or provide P-value in the results section.

4. Line 209 to 210: Consider toning this sentence down. Recent studies have shown aposematic colorations may have important implications in anuran species diversification and genetic differentiation (below papers).

Arbuckle, K. and Speed, M.P., 2015. Antipredator defenses predict diversification rates. *Proceedings of the National Academy of Sciences*, 112(44), pp.13597-13602.

Medina, I., Dong, C., Marquez, R., Perez, D., Wang, I.J. and Stuart-Fox, D., 2023. Anti-predator defenses are linked with high levels of genetic differentiation in frogs.

5. Line 498 onwards: Anurans are known to change their color to regulate their body temperature (e.g., by phenotypic plasticity). The present approach employed by the authors does not accurately account for this. I suggest authors acknowledge this limitation in the methods.

Alho, J.S., Herczeg, G., Söderman, F., Laurila, A., Jönsson, K.I. and Merilä, J., 2010. Increasing melanism along a latitudinal gradient in a widespread amphibian: local adaptation, ontogenic or environmental plasticity?. *BMC Evolutionary Biology*, 10(1), pp.1-9.

Tattersall, G.J., Eterovick, P.C. and de Andrade, D.V., 2006. Tribute to RG Boutilier: skin colour and body temperature changes in basking *Bokermannohyla alvarengai* (Bokermann 1956). *Journal of Experimental Biology*, 209(7), pp.1185-1196.

6. Line 618: I may have missed it. Please add details about the type of analysis performed and the predictors. Was this a PGLS or GAM? Provide the same details in the Extended Data Fig.2 legends. I understand this is possibly the only global compilation of chytridiomycosis severity data available. But I would like to bring to the author's attention the problems in this dataset (see below letter to the original study). <https://www.science.org/doi/full/10.1126/science.aay1838>.

7. Line 621: Should 'the species affected by' must be 'the species UNAFFECTED by'? The current sentence is a bit confusing.

8. Line 717: Here, the n is mentioned as 1,291. But in the methods, the number of species with chytridiomycosis data is described as 258. Please provide full details on the number of species affected and unaffected by chytridiomycosis in the methods section.

9. Line 740: Extended Data Fig 4. Consider adding details about the black line (1:1 line?).

REVIEWER COMMENTS

Reviewer #1 (Remarks to the Author):

Comment: I thank the authors for their revision and for taking into consideration my previous suggestions. While the manuscript has improved, I do still have some concerns/suggestions as detailed below.

Species-level analysis vs assemblage-level analyses:

In my previous review I suggested to make this level of analysis the main level of analysis simply because this is closer to the level at which selection is happening. The authors counter this suggestion by stating in their response letter that:

(1) assemblage-level analyses are more relevant to their questions because “biogeographical patterns, differences in the relative importance of drivers among the biogeographical realms and the colour lightness diversity are purely spatial”. Given the high levels of endemism in anurans, it is eminently possible to carry out realm-specific analyses at the species level (just include all species found in each realm, even if there may be a few cases where the same species pops up in different realms). While it is true that lightness diversity can only be studied at the level of the assemblage, these results are not particularly prominent nor relevant to the main message of the manuscript. As a matter of fact, I could not find any reference to lightness diversity in the Discussion. I suggest that they could be easily shifted to the suppl. material or dropped from the manuscript altogether.

Response: We agree that the high level of endemism may technically allow differentiation of the responses between realms if we accept a certain degree of inflation of species-realm combinations in the data (our solution to this problem is shown in Extended Data Fig. 2). However, because both predictors (the environmental data and biogeographical realm) in this specific analysis are spatially explicit, an assemblage-level approach is more appropriate and requires no additional thresholds or assumptions about the realm association. Note that in your previous response, we mention several general reasons alongside this specific example. Our rationale is that differences among lineages (e.g. families) should be tested at the species-level whenever traits are used as the main predictors. When mainly spatial predictors (e.g. realms or environment only) are used, the analysis should be spatially explicit; this is at the assemblage-level. However, for trait-environment interactions with no additional predictors, we provide both species- and assemblage-level results. Our approach is appropriate for the objective to study how environmental factors structure communities through physiological mechanisms (L74), while in the context of trait evolution or trait-trait interactions, species-level approaches might be more suitable. For instance, we analyse the links between colour lightness and activity or pathogen severity at the species-level (colour, severity, and activity are species traits). Together with the Extended Data and the Supplement, most illustrative items and analyses are at the species-level. Regarding the prominence of displaying the illustrative items, we have focused on assemblage-level results because this was the main objective of our study. Note that the PGLS results are only presented in the text because the few estimates do not qualify as a separate table or coefficient plot. Please let us know if a specific and relevant model or test is missing. We see no technical or theoretical reason why our main scope is misleading or incorrect, but included all requested analysis. We are convinced that your suggestions on extending the species-level perspective enriched and strengthened the article and hope that you find the article useful and comprehensive, as well as that, despite our community focus, it also provides relevant insights for your research.

We accidentally referenced the wrong result in the discussion of colour lightness diversity. In lines 257-259 and lines 275-277, we now include the correct reference but also moved the results of the colour lightness diversity analysis to the Extended Data.

Comment: (2) “data was quantified from literature sources with a geographical scope, so that the strength of our data is the high spatial coverage rather than the species coverage”. I do not quite understand why this is the case, all estimates of colour were averaged obtaining a single value per species, thus there is no spatial component to coloration, other than the distribution of each species.

In sum, I am not convinced that shifting the focus from assemblage to species-level analyses would be so disruptive or change the scope of the paper. After all, when you discuss the implications of your research these refer to the capacity of species to cope with changes.

Response: From an evolutionary standpoint, all biological patterns are shaped by adaptations. However, to assess the extent to which community assembly processes selectively influence certain species within the species pool, assemblage-level approaches are the most appropriate (see more arguments below). Trait-based community ecology underscores that the functional composition of assemblages depends on the spatial context. The contemporary distribution of species mainly reflects the interplay between traits and spatial factors such as climate and biogeography that filter species based on their traits.

For example, if colour lightness provides a thermoregulatory advantage in cold regions but not in warm regions, then describing the spatial variation in colour lightness solely in terms of its relationship with the mean temperature across a species' range would be insufficient. The broader the environmental niche of a species, the less effective central tendencies are in describing gradients. Assemblage-level analysis allows us to discern and differentiate such patterns. In colder regions, most co-occurring species may exhibit darker colours, but not all dark-coloured species are restricted to, or exclusively found in, colder regions.

When investigating the environmental drivers of global trait variation, the assemblage approach has been the primary level of analysis since the inception of community ecology two decades ago. We tested our main hypothesis that environmental factors drive colour lightness variation through physiological mechanisms at both the species and assemblage levels. We believe that this comprehensive way of documentation is of benefit for a broad readership.

Advantages of assemblage-level approach

Accounting for trait variation within communities: Assemblage-level approaches allow to account for the full range of trait variation within communities, which includes multiple species with different trait values (Cornwell and Ackerly 2009).

Addressing community-level responses: Assemblage-level approaches are better suited to investigate how communities collectively respond to environmental changes, making them useful for assessing ecosystem functioning and stability (Loreau et al. 2001).

Overcoming taxonomic limitations: Assemblage-level approaches are less constrained by taxonomic limitations, making them more adaptable for studying diverse and complex ecosystems with numerous species (Cadotte et al. 2011)

Identifying emergent patterns: Assemblage-level approaches can reveal emergent patterns and relationships that may not be apparent when focusing solely on individual species, facilitating the discovery of community-level ecological principles (Flynn et al. 2009).

Cornwell, W. K., & Ackerly, D. D. (2009). Community assembly and shifts in plant trait distributions across an environmental gradient in coastal California. *Ecology*, 90(10), 3424-3431.

Loreau, M., et al. (2001). Biodiversity and ecosystem functioning: current knowledge and future challenges. *Science*, 294(5543), 804-808.

Cadotte, M. W., Carscadden, K., & Mirotnick, N. (2011). Beyond species: functional diversity and the maintenance of ecological processes and services. *Ecology Letters*, 12(10), 1079-1092.

Flynn, D. F., Mirotnick, N., Jain, M., Palmer, M. I., & Naeem, S. (2011). Functional and phylogenetic diversity as predictors of biodiversity–ecosystem-function relationships. *Ecology*, 92(8), 1573-1581.

Comment: In addition, currently the species-level analysis focuses on presenting results at the family-level, but the reason for this is unclear to me: For example, in Ln 175 the authors argue that given the phylogenetic effects this “therefore” implies that estimates need to be computed separately for each family. I cannot follow the logic here. While such analyses may allow us to see whether there are some differences between taxonomic groups, I would like to see an overall species-level analysis that would include all species for which there is data (now you exclude families with fewer than 10 species). Please include a simple model with main effects only, so that these results can be compared with that of other studies. Family-specific effects can be moved to the online material.

Response: We added the results of PGLS models for all species (L175-178). As you can see the model results suggest only a weak general relationship of colour lightness with environmental predictors. By contrast the analysis that differentiates this analysis for families shows strong support for certain mechanisms in certain families. Hence pooling all species and assuming a phylogenetically independent response obscured all relevant patterns.

Comment: It still remains unclear to me how phylogenetic relatedness was included in the species-level analyses. Was this done using Lynch’s method? Please clarify and provide full results in online suppl. table. I suggest that a simple pglis with lambda correction may be the most suitable alternative here.

Links between pathogen-resistance and colour:

The authors now include a very interesting comparative analysis testing for an association between lightness and resistance to fungal infection (arguably the main pathogenic threat to anurans). However, more information on how these analyses were carried out is needed, as the information in Ln 618-623 is too limited.

Response: Thank you for highlighting these points and in general for providing the inspiration for this very interesting analysis. Specific points are addressed below.

Comment: (a) How did you define co-occurring species?

Response: Co-occurring species are species that occur in the same grid cell as the species for which we had severity data from Scheele et al. 2019.

Comment: (b) What was the response variable in the analysis (to find this one has to dive into the extended data)?

Response: The response variable was the interaction of productivity (Annual EVI) and colour lightness for each family. We now clarify this in lines 189-191.

Comment: (c) Why only include productivity of all climatic predictors?

Response: We include the interaction of productivity and colour (lighter coloured species in productive environments) as a proximal mechanistic driver of chytridiomycosis severity. We are unaware of a theoretical foundation for assuming an effect of other mechanisms such as temperature in interaction with colour lightness on chytridiomycosis severity and hence do not think that their inclusion is sufficiently justified.

Comment: (d) How well documented is the link between productivity and pathogen pressure assumed here?

Response: It has been shown that the pathogen load is higher in tropical regions, as the lack of seasonal variation leads to stable environmental conditions and thus stable parasite abundance and thus higher pathogen pressure in tropical environments (Møller, 1998). Hochberg and van Baalen (1998) developed a co-evolutionary model and found that parasite-host interactions evolve along a productivity gradient. This model predicts that in regions with higher productivity, the investment in host defense mechanisms against parasites is the highest. This can be linked to the assumption made by Janzen (1970) and Cornell (1971) that parasites are more prevalent and have a greater impact on hosts in tropical regions compared to other climates. Our motivation for this analysis was to test this specifically for anurans and a well-documented pathogen.

Møller, AP. Evidence of Larger Impact of Parasites on Hosts in the Tropics: Investment in Immune Function within and outside the Tropics, *Oikos*, **82**, 265–270 (1998)

Janzen, D. H. Herbivores and the number of tree species in tropical forests. *Am. Nat.* **104**, 501–528 (1970)

Connell, J. H. On the role of natural enemies in preventing competitive exclusion in some marine animals and in rain forest trees. - In: den Boer, P. J. and Gradwell, G. R. (eds), *Dynamics of numbers in populations*. Centre for Agricultural Publication and Documentation, Wageningen, 298–312 (1971)

Comment: (e) Again, families with <10 species are excluded, why? Given the limited sample size you should strive to include as many species with data as possible. I do not see the logic to run the main analysis separately per family.

Response: We now also provide the results of a PGLS model for all species. Comparing the overall and the family-level model you can see that because the colour lightness of families follows different prevalent drivers the overall signal is obscured and the PGLS model simply indicates that there is no overall effect when accounting for phylogenetic relationships

(the effect of drivers hence depends on the phylogenetic lineage – e.g. family). Groups with less than 10 data points typically do not show significant effects simply because of poor sampling/data availability. Excluding those groups is a common threshold that improves the robustness of statistical tests (see also Delhey et al. 2021).

Comment: (f) Why is the interaction between productivity and lightness the relevant test here? Please add analyses without interaction term as tables to the suppl. material as well.

Response: This interaction addresses a specific mechanism that is theoretically well supported to influence pathogen severity. This is related to question c) as we simply lack a theoretical foundation/expectation for assuming a general influence of colour or that of other mechanisms such as thermoregulation (darker where colder: temperature in interaction with colour). We added the requested test for separate effects of colour or productivity to the supplement.

Comment: (g) What kind of model was used for this analysis, Lynch's method? Please clarify. As above, I suggest that a simple pglis with lambda correction may be the most suitable alternative here.

Response: We did not use the P or S component but the 'raw' colour lightness for this analysis. As for question e): the effect of the driver depends on the family. Accounting for phylogenetic relationship in PGLS analysis reveals a weak general effect. Although only 9 of 20 families show this effect of colour-based pathogen protection on the susceptibility to chytridiomycosis we are convinced that the results are ecologically relevant and motivate further investigations.

Comment: Presentation of results:

I find that the main statistical results as presented in Fig2b, Fig 3b and Fig 4 are very hard to follow. Effects for all predictors are shown on the same graph making it very difficult to recognize overall patterns. One possibility for improvement would be to present forest plots for each predictor in a separate panel.

Response: Thank you for this suggestion. We tried to produce a forest plot for Figure 4 accordingly, but it induced a lot of white space and redundancies, which should be avoided according to the journal guideline for figure preparation. We agree that it is a lot of information for one plot, but the model results cannot be separated as the model needs to contain all four predictors and the confidence intervals for their effects require each forest plot to have a rather large width. As shown in the last revision a table would be even more spacy.

Comment: In lines 140-142 you state "Hence, accounting for idiosyncrasies of the responses among realms with different biogeographical histories largely removed the effect of latent spatially autocorrelated variables" Why is this the case? What evidence was used to support this statement?

Response: The model including a spline based smooth term of geographical coordinates to account for spatially structured latent environmental predictors (gam) explains a similar proportion of variance as the model without this term but including an interaction term that

differentiates the effect for biogeographical realm. This is highlighted by the results shown in Fig. 2 that confirm strong differences in the relative importance of environmental drivers among biogeographical realms. Being structured by latitudinal and longitudinal variation as well as broad scale continental geometry, the trends surface term largely reflects these differences.

Reviewer #2 (Remarks to the Author):

Comment: General comments

I have now gone through the revised manuscript. The authors have done a wonderful job of responding constructively to the suggestions both reviewers provided. In the revised version, the authors have broadened the introduction, included an analysis of possible links between color lightness and pathogen resistance, as well as a path analysis testing the role of tree cover on lightness. The authors also performed a separate analysis comparing color lightness measured by two methods (color wheel vs. pixel approach). This new set of analyses strengthens the author's approach and their previous conclusions that anuran color lightness is primarily associated with four main drivers – temperature, elevation, productivity, and UVB radiation. I have a list of minor suggestions below (some are stylistic) that authors may want to consider. Nice work!

Response: Thank you so much!

Comment: Specific comments

1. Line 87 to 96: This entire paragraph in the introduction has a bunch of results and inferences before the actual results! I could not locate the reviewer's comments requesting such an addition to the introduction. I strongly feel that it doesn't belong here. Consider moving it to the discussion (see another alternative below).

Response: During the last revision we added this paragraph as suggested by the journal guidelines. However, we agree that it does a better job at the start of the discussion, and we have now shifted it there.

Comment: 2. Line 99 onwards: Consider moving this new first paragraph of results to the introduction (as the last paragraph). This nicely provides background and predictions about the study.

If the authors decide to keep the texts in Lines 87-96, then these texts (99-104) can be moved before the part that provides brief results and significance of the study (currently lines 87-96).

Response: Thank you for this suggestion, we agree that this paragraph integrates well into the introduction, and we now have shifted it there.

Comment: 3. Line 188: Thanks for this new addition – very interesting results. The authors present standardized effect sizes of the interaction between color lightness and productivity on chytridiomycosis severity (ED Fig.2). It is important to note that only 9 out of 20 families show a positive effect. It is unclear about the effect sizes and significance of the interaction term in the global model. I suggest authors provide the full model results as a table in the supplement or provide P-value in the results section.

Response: Thank you very much! In the results we now mention that severity of Chytridiomycosis was higher for lighter coloured species in regions with a higher productivity but only for 9 out of 20 families (L188). We also added significance levels to Extended Data Fig. 2 and provide the results for all families in the supplement.

Comment: 4. Line 209 to 210: Consider toning this sentence down. Recent studies have shown aposematic colorations may have important implications in anuran species diversification and genetic differentiation (papers below).

Arbuckle, K. and Speed, M.P., 2015. Antipredator defenses predict diversification rates. *Proceedings of the National Academy of Sciences*, 112(44), pp.13597-13602.

Medina, I., Dong, C., Marquez, R., Perez, D., Wang, I.J. and Stuart-Fox, D., 2023. Anti-predator defenses are linked with high levels of genetic differentiation in frogs.

Response: We now included this sentence in our manuscript (L217-219).

Comment: 5. Line 498 onwards: Anurans are known to change their color to regulate their body temperature (e.g., by phenotypic plasticity). The present approach employed by the authors does not accurately account for this. I suggest authors acknowledge this limitation in the methods.

Alho, J.S., Herczeg, G., Söderman, F., Laurila, A., Jönsson, K.I. and Merilä, J., 2010. Increasing melanism along a latitudinal gradient in a widespread amphibian: local adaptation, ontogenic or environmental plasticity?. *BMC Evolutionary Biology*, 10(1), pp.1-9.

Tattersall, G.J., Eterovick, P.C. and de Andrade, D.V., 2006. Tribute to RG Boutilier: skin colour and body temperature changes in basking *Bokermannohyla alvarengai* (Bokermann 1956). *Journal of Experimental Biology*, 209(7), pp.1185-1196.

Response: We now acknowledged this limitation in the method section in lines 484-486.

Comment: 6. Line 618: I may have missed it. Please add details about the type of analysis performed and the predictors. Was this a PGLS or GAM? Provide the same details in the Extended Data Fig.2 legends. I understand this is possibly the only global compilation of chytridiomycosis severity data available. But I would like to bring to the author's attention the problems in this dataset (see below letter to the original study). <https://www.science.org/doi/full/10.1126/science.aay1838>.

Response: We now clarify that we conducted a single regression (OLS) of the interaction between colour lightness and productivity (Annual EVI) on chytridiomycosis severity for each family. Thank you for pointing at the debate around this dataset. Lambert et al. 2020 wrote a comment about the study from Scheele where they found insufficient evidence implicating chytridiomycosis. Scheeles responded that Lambert et al. did not incorporate expert knowledge and that they only used easily accessible sources. We decided to use the more complete data which could have inflated the Type II error in our analysis.

Comment: 7. Line 621: Should 'the species affected by' must be 'the species UNAFFECTED by'? The current sentence is a bit confusing.

Response: We added more detail to clarify this statement (L631-633).

Comment: 8. Line 717: Here, the n is mentioned as 1,291. But in the methods, the number of species with chytridiomycosis data is described as 258. Please provide full details on the number of species affected and unaffected by chytridiomycosis in the methods section.

Response: We have added more detail concerning the chytridiomycosis dataset. Our total dataset includes 1,291 species, 258 from Scheele et al. 2019 and 1033 species that were classified as unaffected because they occurred in the same grid as the affected species.

Comment: 9. Line 740: Extended Data Fig 4. Consider adding details about the black line (1:1 line?).

Response: Thank you for highlighting this. We now clarify that these are the 1:1 lines.

REVIEWERS' COMMENTS

Reviewer #1 (Remarks to the Author):

The authors have carried out the additional analyses I suggested, specifically the climatic effects on anuran lightness and on the severity of fungal infections at the species level. While I do not fully agree with their emphasis on assemblage level analyses, what matters is that both are reported.

I do however strongly suggest to mention in the discussion the fact that species level analyses of the entire sample show substantially weaker and often not statistically significant effects, when contrasted with the much stronger effects found at the assemblage level. Otherwise, many readers may overlook these important discrepancies.

The fact that climatic effects on the severity of fungal infections is much weaker if all species are analysed together should also warrant mention in the discussion.

Ln 178 what does the 'overall p-value' refer to? I suggest to add full statistical details for each effect, slope, SE, t-values df, p-values, etc).

Ln 178-182 "Using the latitudinal distribution centre of each family (absolute values averaged across all species) showed that the relative importance of temperature and UVB increased with increasing latitude, while the importance of productivity showed a U-shaped relationship with a flatter end at low latitudes (Extended Data Fig. 2)." As far as I can see the figure shows species level scatterplots between species-specific climate values and colour. I do not understand what "using the latitudinal distribution centre of each family" means here. Is each data point a family or a species? Is it possible that the suppl. material is not the updated version?

REVIEWERS' COMMENTS

Reviewer #1 (Remarks to the Author):

Comment: The authors have carried out the additional analyses I suggested, specifically the climatic effects on anuran lightness and on the severity of fungal infections at the species level. While I do not fully agree with their emphasis on assemblage level analyses, what matters is that both are reported.

I do however strongly suggest to mention in the discussion the fact that species level analyses of the entire sample show substantially weaker and often not statistically significant effects, when contrasted with the much stronger effects found at the assemblage level. Otherwise, many readers may overlook these important discrepancies.

Response: We now mention and contextualise this result in the discussion (L260-261) as well as clarify which interpretations reference species-level and assemblage-level results.

Comment: The fact that climatic effects on the severity of fungal infections is much weaker if all species are analysed together should also warrant mention in the discussion.

Response: We now mention and contextualise this result in the discussion (L288-291).

Comment: Ln 178 what does the 'overall p-value' refer to? I suggest to add full statistical details for each effect, slope, SE, t-values df, p-values, etc).

Response: Thank you for pointing at this mistake.

Comment: Ln 178-182 "Using the latitudinal distribution centre of each family (absolute values averaged across all species) showed that the relative importance of temperature and UVB increased with increasing latitude, while the importance of productivity showed a U-shaped relationship with a flatter end at low latitudes (Extended Data Fig. 2)." As far as I can see the figure shows species level scatterplots between species-specific climate values and colour. I do not understand what "using the latitudinal distribution centre of each family" means here. Is each data point a family or a species? Is it possible that the suppl. material is not the updated version?

Response: During the revision the former Extended Data Figure 2 became Supplementary Figure 4. Extended Data Figure 2 shows the independent contribution of environmental variables against the latitudinal distribution centre of each family.